# Measuring Meta-Cultural Competency: A Spectral Framework for LLM Knowledge Structures

Sougata Saha [* 1]  Madhur Jindal [* 1]  Saurabh Kumar Pandey [1]  Mahardika Krisna Ihsani [1]  Alham Fikri Aji [1]
Monojit Choudhury [1]

## Abstract

Most existing cultural evaluation frameworks for large language models (LLMs) focus on matching model outputs to ground-truth answers, primarily measuring factual cultural awareness. This overlooks whether models internalize broader cultural structure and pluralism. We introduce a spectral-analysis-based framework that captures large-scale macrostructural patterns in models' cultural knowledge and evaluate eight LLMs across nine cultural domains spanning all five of Newmark's cultural dimensions and 170 countries. Comparing with human data, we find that instruction-tuned models align more closely with human cultural structure than older models, while increased model size does not consistently improve performance. Finally, simulation-based experiments show that our proposed spectral metric better predicts a model's ability to serve users from unfamiliar cultural backgrounds than existing ones.

## 1. Introduction

*"And so these men of Indostan disputed loud and long, each in his own opinion exceeding stiff and strong. Though each was partly in the right, and all were in the wrong!"*–Saxe (1871).

The parable of the blind men and the elephant illustrates the limitations of partial observation: each observer perceives something correct in isolation, but none captures the larger whole. Current LLM evaluation frameworks exhibit a similar limitation in the cultural domain. Most approaches assess cultural awareness by comparing model outputs to ground-truth answers within narrow, localized contexts (Nadeem et al., 2021; Yin et al., 2022; Jha et al., 2023). While informative, current benchmarks reduce cultural knowledge to discrete facts (e.g., *Biryani* is a common food in India or similarities between Indian and Pakistani cuisines), capturing only *microstructural properties* of culture. They fail to capture the *macrostructure*, that is, the broader, recurring patterns, such as how many overarching categories of cuisines exist, whether they are hierarchically structured or randomly distributed, and how much variation occurs within and across national cuisines (Sorensen et al., 2024; Strauss, 1992). This broader capacity, often referred to as *variational awareness*, constitutes a critical dimension of *meta-cultural competency* (Leung et al., 2013; Sharifian, 2013) - the innate human ability to recognize, interpret, and navigate cultural variation across levels of familiarity, and is essential for making sense of both familiar and unfamiliar cultural contexts.

Unlike databases, which store and enable the retrieval of point facts, LLMs are powerful compression engines that structure knowledge in ways that allow them to generate appropriate responses across diverse cultural domains and settings (Talmor et al., 2020; Geva et al., 2021; Pan et al., 2026). Hence, analyzing macrostructures provides a more holistic measure of *cultural awareness*, or as we shall argue, of *variational awareness*, and eventually of *meta-cultural competency*. Macrostructural evaluation also has pragmatic advantages: global structural patterns are easier and more reliable to elicit from humans than idiosyncratic preferences (Triandis, 1989; Shweder, 1991; Matsumoto, 2007). For instance, some domains, such as currency, are highly country-specific, while others, like house numbering[2], are more cross-nationally uniform. Testing whether LLMs capture such relative patterns directly probes their ability to model cultural structure and variation, a ground truth that is tractable and human-verifiable. Yet, despite this promise, macrostructures remain underexplored in LLM evaluation.

We address this gap by introducing a spectral-analysis-based framework (Klema & Laub, 1980; Wall et al., 2003; Stoica et al., 2005; Abdi, 2007) for evaluating the macrostructures

---

[1]Department of NLP, Mohamed bin Zayed University of Artificial Intelligence, Abu Dhabi, UAE. Correspondence to: Monojit Choudhury <monojit.choudhury@mbzuai.ac.ae>.

*Proceedings of the 43rd International Conference on Machine Learning*, Seoul, South Korea. PMLR 306, 2026. Copyright 2026 by the author(s).

---

[2]Although house numbering conventions may vary, the numerals themselves are mostly universal.

of cultural knowledge in LLMs. Using this framework, we analyze eight models of varying sizes across nine cultural domains (e.g., food, religion, language, currency, and holidays) that collectively span all five cultural dimensions in Newmark's taxonomy (Newmark, 1988), covering 170 countries. This macrostructural evaluation complements existing microstructural benchmarks by revealing large-scale cultural patterns not captured by fact-based probes. Our contributions are as follows:

1. **We introduce a spectral-analysis-based framework** that shifts cultural evaluation from microstructural probing to domain-level macrostructural modeling, enabling principled assessment of variational awareness as a foundation for meta-cultural competence.

2. Using data from 170 countries across nine cultural domains, **we present the first large-scale macrostructural analysis** of eight LLMs of varying sizes, showing that such evaluation is tractable, interpretable, and aligned with human cultural structure.

3. Through simulation-based experiments, **we show that the proposed spectral metric better predicts a model's ability to serve users** from unfamiliar cultural backgrounds than existing benchmarks, capturing higher-order *explication strategies* (Sharifian, 2013) expected of meta-culturally competent AI systems[3].

## 2. Defining Culture

We ground our evaluation framework in theories from the social sciences that provide essential conceptual grounding for how cultural knowledge is structured, shared, and navigated across contexts.

**Structural Organization of Cultural Knowledge:** Cultural knowledge can be understood at complementary levels of organization. *Microstructures* capture localized factual associations, such as knowing that the currency of Japan is the Yen. In contrast, *macrostructures* reflect the global organization of knowledge across domains, such as how currencies distribute across countries or how cuisines cluster geographically. This distinction mirrors other fields: in physics, microstates describe particle-level configurations while macrostates capture emergent system properties (Reif, 2009; Huang, 2008); in cognitive science, higher-level structures enable abstraction beyond local facts (Carey, 2000; Lake et al., 2017; Kemp & Tenenbaum, 2008); and in anthropology, recurring cultural patterns emerge from localized practices (Lévi-Strauss, 1963). As Anderson (1972) famously argued, "more is different," emphasizing that higher-order structure is not reducible to isolated facts. Conse-

quently, evaluating cultural knowledge requires attention to both micro- and macro-level representations.

From a computational perspective, we adopt a structural view of cultural knowledge in LLMs, modeling it as a network in which nodes correspond to concepts and edges encode associations between them (Tenenbaum et al., 2011; Nickel & Kiela, 2017). In this view, existing evaluation methods (Appendix A.8) primarily probe *microstructures* by testing localized portions of this network, such as first- or $k^{th}$-order associations for small values of $k$ ($\ll n$, the number of nodes), and validating them against curated lists. While informative, such probes capture only pointwise relationships in small subgraphs and do not reveal whether models internalize broader organizational principles, motivating the need for *macrostructural* evaluation.

**Cultural Consensus and Macrostructure:** A foundational framework that operationalizes shared cultural structure is *Cultural Consensus Theory (CCT)* (Romney et al., 1987; Weller, 2007). CCT infers collective knowledge by analyzing patterns of agreement across individuals, assuming that when a shared cultural model exists, disagreement reflects variation in knowledge rather than objective truth. It identifies dominant shared views from these agreement patterns and uses factor loadings to estimate how strongly individuals align with them, without requiring predefined ground truth. We draw on this perspective to motivate macrostructural evaluation: rather than assessing isolated facts, we analyze whether LLMs recover the same large-scale structures observed in the respective cultural domains.

**Meta-Cultural Competency:** Building on this structural understanding of culture, *meta-cultural competency* refers to the ability to reason about cultural variation itself. It enables individuals to communicate and adapt across cultural contexts (Sharifian, 2013) and consists of three components: **(i)** *Variational awareness* **(VA)** refers to recognizing that cultural practices, beliefs, and preferences vary across groups, and that one's own knowledge may be incomplete. **(ii)** *Explication strategies* enable actors to act on this awareness by asking clarifying questions or making assumptions explicit in unfamiliar settings. **(iii)** *Negotiation strategies*[4] support the joint resolution of misunderstandings through interaction. For instance, a foreign traveler in Japan may notice unfamiliar dining customs (VA), ask whether it is customary to say *"Itadakimasu"* before eating (explication), and adapt their behavior accordingly (negotiation).

Saha et al. (2025) argues that meta-cultural competency is a necessary system-level ability for cross-cultural AI. We extend this framework by proposing **macrostructural evaluation as a measurable proxy for VA**. If a model

---

[3]Code and data available at https://github.com/mbzuai-nlp/metaculturemacro.

[4]Included for conceptual completeness; not experimentally evaluated in this study.

captures human-like cultural macrostructures, it is better positioned to recognize and respond to cultural variation. In later sections, we empirically show that macrostructural alignment predicts a model's ability to serve users from unfamiliar cultural backgrounds.

# 3. Approach

Here we formally define and evaluate our framework.

## 3.1. Formal Definition

Following Adilazuarda et al. (2024), we define $D = \{d_1, \ldots, d_m\}$ as a set of $m$ *semantic proxies* or cultural domains (e.g., food, holidays, religion), and $C = \{c_1, \ldots, c_n\}$ as a set of $n$ *demographic proxies* that serve as observable demographic attributes over which cultural variation is observed. In this work, we use *countries* as proxies, as they are among the most widely available and consistently defined demographic proxy (Adilazuarda et al., 2024).

Each domain $d_i$ is associated with a set of $k$ semantically similar questions $Q^{d_i} = \{q_1, \ldots, q_k\}$ posed over a set of $t$ items $I^{d_i} = \{i_1, \ldots, i_t\}$. For example, in the currency domain, $I^{d_i}$ may correspond to the set of all world currencies. Let $M_\theta$ denote a model parameterized by $\theta$. For each country $c \in C$, the model answers a domain-specific question $q_j \in Q^{d_i}$ by producing a probability distribution $\{p_{c,i}^j\}$ over the $t$ items, where $\sum_{i=1}^{t} p_{c,i}^j = 1$. This distribution captures the model's *microstructural knowledge*: its probabilistic associations between proxies and domain items. Existing cultural evaluations typically compare $p_c^j$ against ground-truth distributions. Here, we extend this formulation to capture *macrostructural* knowledge.

For each domain $d_i$, we aggregate country-level distributions across all $n$ countries to form an $n \times t$ matrix $H_{c,i}^j = p_{c,i}^j$. We construct an adjacency matrix $A^{d_i} \in \mathbb{R}^{n \times n}$, where each entry encodes the cosine similarity between the distributions of a pair of countries:

$$A^{d_i} = D^{-1} H^j H^{j\top} D^{-1} \qquad (1)$$

$$D = \text{diag}(\|p_{c_1}^j\|_2, ..., \|p_{c_n}^j\|_2) \in \mathbb{R}^{n \times n} \qquad (2)$$

We analyze $A^{d_i}$ using two spectral metrics: (i) The entropy-based **Effective Rank (ER)**, which measures the number of significant dimensions of $A^{d_i}$ (Roy & Vetterli, 2007). ER $\approx 1$ when dominated by one eigenvalue, and ER $\approx n$ when eigenvalues are uniform. (ii) Inspired by CCT, **Spectral Gap Ratio (SR)** captures the gap between the first and second eigenvalues. A high SR indicates strong cross-country similarity, while a low SR reflects country-specific variation. Let $\{\lambda_1, ..., \lambda_n\}$ denote all eigenvalues of $A^{d_i}$ in descending order and $\tilde{\lambda}_i = \lambda_i / \sum_{j=1}^{n} \lambda_j$, where $\lambda_i > 0$,

then

$$\text{ER} = \exp(-\sum_{i=1}^{n} \tilde{\lambda}_i \log \tilde{\lambda}_i), \text{ and } \text{SR} = \frac{\lambda_1}{\lambda_2} \qquad (3)$$

### 3.1.1. MACROSTRUCTURAL REGIMES OF CULTURAL DOMAINS

The ER and SR metrics capture complementary aspects of macrostructure: ER reflects the diversity or pluralism of patterns within a domain, whereas SR reflects the degree of shared structure across countries. Together, they induce four interpretable macrostructural regimes:

**(i) Low ER, High SR (LH):** A small number of dominant patterns that are widely shared across countries, corresponding to a near-universal consensus. Structurally, this resembles a dense or fully connected graph. Domains such as house numbers fall into this regime, as their distributions are largely uniform across countries (Mukherjee et al., 2024).

**(ii) High ER, Low SR (HL):** Many distinct patterns with little cross-country similarity, yielding a highly fragmented structure. This regime resembles disconnected or random graphs (Bollobás & Bollobás, 1998), and is characteristic of domains such as currency, where each country largely follows its own system.

**(iii) High ER, High SR (HH):** High diversity coupled with shared structure, where countries form affinity clusters. This regime resembles small-world or core-periphery structures (Newman, 2000). Food-related domains exemplify this case, where global staples coexist with strong regional variations.

**(iv) Low ER, Low SR (LL):** Few dominant patterns, but weak global consensus, resulting in multiple distinct clusters rather than a single shared core. Religion illustrates this regime, with a limited number of major belief systems but clear geopolitical clustering.

Figures 1 (top) and 6 show exemplary t-SNE visualizations for each regime (see Appendix A.3), illustrating how ER and SR jointly describe the global structure. Together, these regimes provide an interpretable space for analyzing how cultural variation and consensus manifest in macrostructures.

### 3.1.2. SPECTRAL CHARACTERIZATION OF VA

Building on the macrostructural regimes defined by ER and SR, we formalize their relationship to VA (*variational awareness*). Let $A^{d_i} \in \mathbb{R}^{n \times n}$ denote the country-country similarity matrix for domain $d_i$, with eigen decomposition $A^{d_i} = U \Lambda U^\top$, where $\Lambda = \text{diag}(\lambda_1, \ldots, \lambda_n)$ and $\lambda_1 \geq \cdots \geq \lambda_n \geq 0$. The spectrum $\{\lambda_i\}$ characterizes how cross-country cultural structure in domain $d_i$ is distributed across latent dimensions, where a dominant leading eigenvalue indicates strong global consensus and a flatter spec-

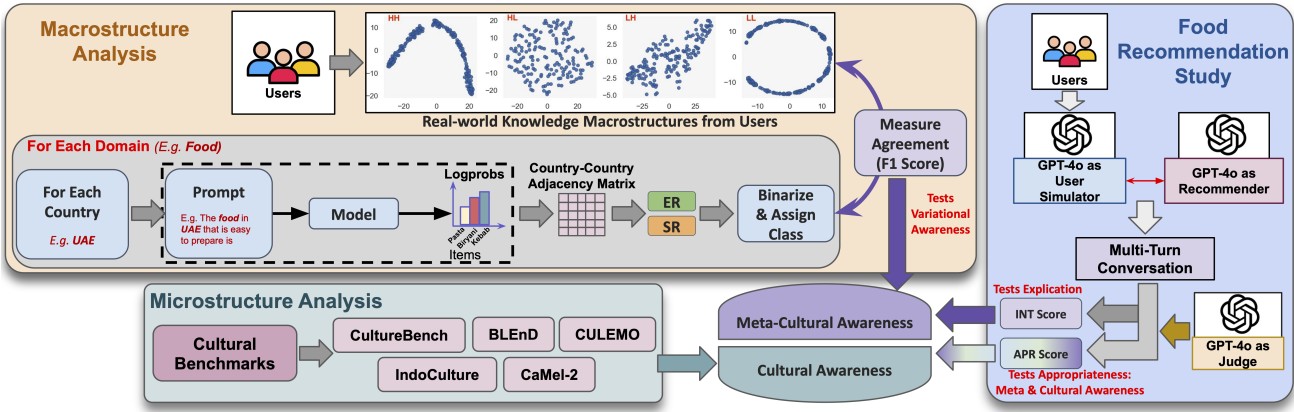

*Figure 1.* Overview of our end-to-end evaluation framework. We assess variational awareness via macrostructural analysis (F1) and explication (INT) through recipe recommendation simulation, with appropriateness (APR) capturing both cultural and meta-cultural awareness. Microstructural benchmarks, shown for comparison, capture factual cultural knowledge and provide complementary signals.

trum indicates multiple competing patterns of variation. For example, in food domains, one dimension might distinguish rice- versus wheat-based cuisines, another fish- versus meat-dominant diets, and so on, each reflecting distinct global patterns of variation. The normalized spectrum $\tilde{\lambda}_i$ (Equation 3) defines a distribution over latent cultural dimensions. For models, let ER and SR jointly define a domain's regime through a mapping $r : (\text{ER}, \text{SR}) \mapsto \{\text{LL}, \text{LH}, \text{HL}, \text{HH}\}$. This mapping $r$ is an operational choice and discussed in Section 3.3. We define VA as a system-level measure of regime agreement with human-collected ground truth $y_{\text{human}}^{d_i}$ across domains, aggregated by $\mathcal{A}(\cdot)$:

$$\text{VA} = \mathcal{A}\Big( \big\{ \mathbb{1}[r(\text{ER}(A_{\text{model}}^{d_i}), \text{SR}(A_{\text{model}}^{d_i})) = y_{\text{human}}^{d_i}] \big\}_{d_i \in D} \Big)$$

Where $\mathbb{1}[\cdot]$ is the indicator function and $\mathcal{A}(\cdot)$ denotes an aggregation function over domains (e.g., macro-F1 or accuracy). Under this definition, VA captures whether a model recognizes the presence or absence of diversity (ER) and consensus (SR) across cultural domains, rather than matching exact spectral values for individual domains. Low VA arises when a model consistently misclassifies regimes by collapsing diverse patterns into a single dominant structure (over-homogenization) or by fragmenting shared structure into noise (over-discretization). This regime-based formulation yields a discrete, interpretable, and domain-agnostic definition of VA, consistent with prior definitions (Saha et al., 2025) and consensus-based views of CCT.

To validate this interpretation, we conducted a human study next across nine cultural domains.

### 3.2. Human Grounding of Macrostructural Regimes

Let $R$ denote a ranking $d_i \geq d_j \geq \cdots \geq d_m$ over domains $D$, ordered by how prevalent and consensus-driven participants expect answers to domain-specific questions

$Q^{d_i}$ to be across countries $C$. To obtain a human-grounded ranking $R$, we surveyed with 80 participants from 16 world regions (5 per region; details in Appendix A.2). Participants were asked to rank nine domain-specific questions, each chosen to concisely capture an observable property of the corresponding cultural domain. After two rounds of pilot testing with 10 internal graduate-level participants, the final survey was hosted on Google Forms and administered via Prolific, which included two attention checks to ensure response quality. Participants were compensated at an hourly rate of £9 and took 5-10 minutes to complete the survey.

Table 1 lists the nine questions and the aggregated rankings obtained by averaging participant responses. Participants generally judged domains such as convenient food and house numbers to exhibit higher cross-country consensus, while domains such as national dish and currency were judged to be more divergent, reflecting higher expected consensus (SR) in the former and greater divergence in the latter. To estimate expected pluralism (ER), three pilot participants independently annotated the likely diversity of items per domain and reached consensus, ensuring shared judgment rather than individual bias. Table 1 (column *Category*) summarizes both ER-SR dimensions, along with the number of items used for LLM evaluation in subsequent experiments.

Together, the nine domains span all five dimensions of culture (material, ecology, social, customs, and habits) in Newmark's taxonomy (Newmark, 1988) and are present across both Western and non-Western societies (Katan & Taibi, 2021). Moreover, they instantiate all four macrostructural regimes (LL, LH, HL, HH), ensuring that our evaluation captures a broad range of cultural configurations. This diversity allows the ER/SR metrics to generalize beyond specific domains by measuring structural properties of consensus and variation that underlie cultural knowledge more broadly.

| # | Domain | Question | Avg Score | Std Dev | Category | # Items |
|---|--------|----------|-----------|---------|----------|---------|
| 1 | House numbers | House numbers in your country | 3.18 | 2.23 | LH | 1000 |
| 2 | Convenient foods | Easy to prepare/buy foods in your country | 2.90 | 2.01 | HH | >3700 |
| 3 | Common foods | Commonly eaten foods in your country | 3.92 | 1.86 | HH | >3700 |
| 4 | Healthy foods | Healthy foods in your country | 3.98 | 1.62 | HH | >3700 |
| 5 | Religions | Major religions practiced in your country | 4.70 | 2.33 | LL | 21 |
| 6 | Holidays | Holidays and festivals celebrated in your country | 5.64 | 1.73 | LL | 2500 |
| 7 | Languages | Most common languages spoken in your country | 5.89 | 2.28 | LL | 161 |
| 8 | National dish | National dish of your country | 6.59 | 2.17 | HL | >3700 |
| 9 | Currency | Currency used in your country | 8.20 | 1.80 | HL | 168 |

*Table 1.* Domain Categorization by Humans

### 3.3. Model Evaluation Setup

We now operationalize the regime-based formulation of VA by estimating ER and SR from model-induced distributions. For each question (Table 1, column *Question*), we curated an exhaustive list of candidate items spanning 170 countries from diverse sources to ensure global coverage and enough in-domain diversity (see Appendix A.1). For every question $q_j \in Q^{d_i}$, we constructed a prompt and varied the country, yielding $m \times k$ prompts across all domains. We extracted models' log probabilities over the full item list, summed log probabilities for multi-token items, and applied a softmax to obtain country-level distributions (see Appendix A.4). This probabilistic evaluation captures uncertainty over low-likelihood associations that decoded outputs may miss.

These distributions were stacked into matrices (Section 3.1) and country-country adjacency matrices of size $170 \times 170$ were computed using Equation 1. For each adjacency matrix, we computed ER and SR (Equation 3). To facilitate categorical comparison with human judgments, ER and SR values were binarized as high or low relative to their medians across all questions and models, and the resulting labels were combined to assign each domain-question pair to one of the four ER-SR regimes. Figure 1 illustrates the end-to-end evaluation pipeline. As discussed in Appendix A.5, this discretization does not alter the underlying geometry of the ER-SR space or its correspondence with human judgments, but provides a consistent boundary for comparison.

We evaluated eight open-weight language models of varying sizes (Table 4). For each of the $m = 9$ domains, we used $k = 3$ questions (27 prompts total; Appendix A.7). All experiments were run using vLLM (Kwon et al., 2023) with FP16 quantization on two NVIDIA RTX 6000 GPUs, totaling approximately 50 GPU hours.

### 4. Results

#### 4.1. Model Alignment with Human Judgments

We evaluate whether LLMs capture human-aligned macrostructural regimes, and thus exhibit variational awareness, by comparing model-assigned ER-SR categories against human judgments (Section 3.2). For each model, we compare model-assigned ER-SR categories with human-derived ground truth data (Section 3.2) and report macro-F1 scores in Figure 2, along with the category-wise scores. We further performed a bootstrap analysis (10,000 resamples) to estimate 95% confidence intervals for the macro-F1 scores, as illustrated in Figure 3 and detailed in Table 5. We observe the following: (i) Llama-8B performs best, followed by Llama-70B and Aya-8B. Interestingly, Llama-70B does not clearly outperform its much smaller parameterized 8B counterpart, suggesting that **macrostructural competence may not scale with model size** as it does on existing benchmarks like MMLU (Hendrycks et al., 2021), SQuAD (Rajpurkar et al., 2016), GSM-8K (Cobbe et al., 2021), etc. (ii) Similarly, Gemma-2-2B and 9B versions perform similarly, leading us to conjecture that **instruction-tuned models may plateau in macrostructural ability beyond a certain size**, which in our case is at 2B parameters. (iii) Older and smaller models, such as GPT-2, perform substantially worse than newer instruction-tuned LLMs and even larger non-instruct models like GPT-J-6B. (iv) Llama-3.2-1B performs above GPT-2 but below GPT-J-6B, supporting the idea that **a certain parameter threshold is necessary before macrostructural performance stabilizes**. (v) Strikingly, **all eight models consistently misclassify food domains as HL rather than HH**, treating national cuisines as disconnected despite humans expecting both diversity and cross-country similarity. This systematic failure across models suggests that current LLMs fail to internalize the relational structure of food culture. **They possibly learn discrete and stereotypical national categories** without regional clustering or shared culinary elements. (vi) Llama-3.2-1B and GPT-J-6B only predict HL and LH, missing nuanced HH/LL categories, whereas GPT-2 always defaults to LH, failing to even acknowledge cultural variations. In Figure 2, **we observe an increase in the number of correctly predicted categories as model complexity increases from fewer parameterized and less complex (left) to more parameterized and more complex (right)**. Although it eventually plateaus, where no model can correctly classify the HH category. We further illustrate a heatmap of the ER and SR values for all models and questions in Figures 9 and

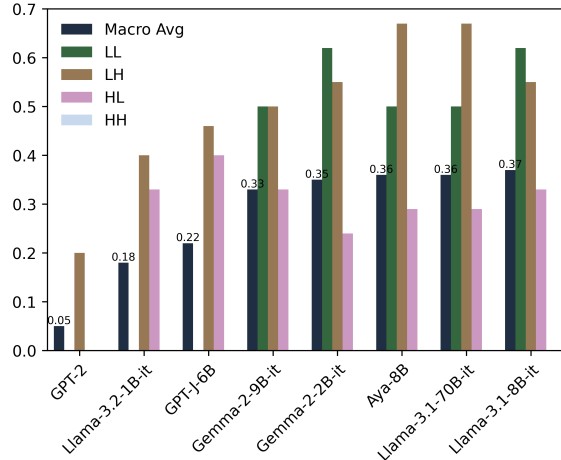

*Figure 2.* Model & Category-wise Macro-F1.

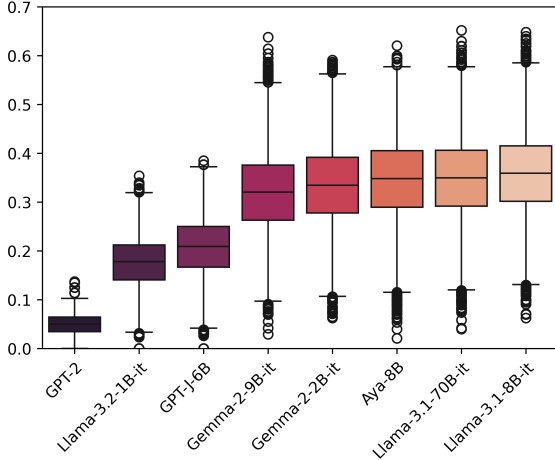

*Figure 3.* Bootstrapped Confidence Intervals.

10, and the per-domain confusion matrices for all models in Figure 11. Figure 12 illustrates the actual t-SNE representations from the adjacency matrix of selected domains across all models, for comparison with the sample t-SNE representations in Figure 1.

## 4.2. Model Rankings: Micro vs. Macrostructural

To compare insights from macrostructural versus microstructural analyses, we examined recent cultural benchmarks published in the last three years and extracted model rankings by domain. We found five benchmarks that overlap with some of our nine domains and eight models. Table 9 lists the benchmarks and the ranking among models from each paper. We observe that Llama-3.1-8B-it is consistently ranked higher than Aya-8B in existing studies, which aligns with our results. However, unlike our results, these studies rank Gemma-2 variants above both Llama and Aya. They also report Llama-70B as performing significantly better than others, whereas our macrostructural analysis suggests that performance asymptotes in instruction-tuned models beyond 2B parameters (Gemma-2-2B-it).

Furthermore, pairwise t-tests between bootstrapped macro-F1 distributions from the macrostructural analysis show that not all model pairs are significantly different ($p < 0.05$). Only GPT-2 is significantly different from other models except Llama-3.1-1B-it. The pairwise differences between other models are not statistically significant, indicating no significant ordering, unlike what is indicated by the microstructures. These discrepancies highlight how macrostructural evaluation reveals different aspects of models' cultural knowledge than microstructural methods, which focus on point-wise factual accuracy.

## 5. Evaluating Meta-Culture via Simulation

We designed a simulation study to test whether macrostructure-based performance predicts meta-cultural competency in downstream tasks. As a case study, we focus on recipe recommendation (Yang et al., 2024; Lyu et al., 2024), a common AI application where cultural knowledge, personalization, and explication are central. Prior work shows that food-related decision-making is inherently interdisciplinary and cross-cultural, engaging multiple dimensions of culture (Newmark, 1988), including nutrition, religion, economy, geography, and collective identity (Abarca, 2006; Khare, 1992; Mintz & Du Bois, 2002; Toledo et al., 2019; Rozin, 2005). Consequently, recipe recommendation integrates national and everyday food practices, religious norms, economic constraints, and habitual routines, requiring context-sensitive reasoning across domains and making it a robust testbed for evaluating meta-cultural competency rather than a narrowly scoped task.

### 5.1. Simulation-Based Evaluation Setup

**User Simulation:** To scale the setup, we replaced human users with an LLM-based simulator. We first collected breakfast-habit data from 10 participants (the same pilot participants in Section 3.2), representing 5 distinct cuisines and a range of dietary profiles: vegetarian, non-vegetarian, halal, diabetic, varying cooking skills, and nutrition goals. Each participant interacted with GPT-5 (ChatGPT) through a structured prompt (in Box 1 and Box 2) that instructed GPT-5 to act as an anthropologist and elicit detailed breakfast-related habits. The participants reviewed and edited the resulting JSON persona summaries to ensure accuracy and authenticity. These finalized personas formed the basis for the GPT-4o user simulator (prompt in Box 3).

**Role-Play Setup:** In the experiment, the user simulator

(GPT-4o) interacted with the six instruction-tuned LLMs from Section 3.3, acting as recipe recommenders (using prompt in Box 5). The recommender was free to ask clarifying questions (as mentioned in the prompt), and the conversation ended once a breakfast recipe was recommended. Both the simulator and recommenders were run with temperature 1, and each scenario was repeated 10 times to yield conversational variations.

**Evaluation:** Conversations were rated by GPT-4o as a judge (using prompts in Box 6 and Box 7) on two dimensions, each from 1 (low) to 10 (high), reflecting meta-cultural competency: **(i) Appropriateness (APR)**: The alignment of the final recommendation with the user persona (preferences, cultural background, dietary restrictions, lifestyle, etc). High scores reflect meta-cultural competency, operationalizing VA through effective explication and culturally appropriate adaptation. **(ii) Interaction (INT)**: The quality of clarifying questions used to elicit user-specific needs, such that it could personalize the recommendation. High scores reflect high explication (operationalizing VA) and instruction following capacity. To perform well, a system must both elicit relevant user preferences through appropriate clarification (captured by the INT score) and generate culturally informed, safe, and personalized recommendations (captured by the APR score).

**Control Setup:** To isolate a model's default capacity of operationalizing its VA via explication, we also ran a control condition where recommenders were not explicitly nudged to ask clarifying questions. We omitted the instruction from the prompt that specified how food preferences are individual-specific and dependent on several personal factors, as illustrated in the prompt in Box 4. Each model engaged in 200 conversations (100 control, 100 experiment), totaling 1200 conversations across six models.

### 5.2. Simulation Results: Meta-Cultural Competency

Figure 4 presents the results of the simulation study, with models on the x-axis in ascending order by their microstructural cultural awareness (Section 4.2). We highlight four main observations: **(i) Llama-3.1-70B-it and Llama-3.1-8B-it perform best.** Both models achieve the highest APR and INT scores. Their performance in the control and treatment groups shows no statistically significant difference, suggesting that they are intrinsically variationally aware and meta-culturally competent to an extent, and already possess strong explication skills without requiring explicit instructions. **(ii) Gemma-2 models perform below Llama variants** ($p < 0.05$), despite showing strong cultural awareness in prior microstructural benchmarks. This drop aligns with their weaker macrostructural rankings (Section 4.1), underscoring that APR captures meta-cultural competency as predicted by macrostructural alignment rather than mi-

crostructural accuracy. Models with low VA cannot effectively explicate and thus fail to achieve high APR scores. Similar to Llama, Gemma shows no significant difference between control and treatment, though Gemma-2B exhibits a small, non-significant APR improvement. **(iii) Aya-8B benefits from nudging.** Unlike Llama and Gemma, Aya shows a statistically significant improvement in the treatment condition, indicating that it does not display meta-cultural competency by default but can be nudged to do so. Interestingly, Aya also outperforms Gemma variants, which is consistent with its macrostructure-based ranking but contrary to its microstructural scores from other benchmarks. **(iv) Llama-3.2-1B-it shows mixed behavior.** Its performance lies between Aya and Gemma and is much lower than other Llama variants. Furthermore, its performance does not change with adding more instructions in the treatment prompt, suggesting a possible lack of following instructions and limited meta-cultural competency.

**Instruction Following vs. Explication:** To test whether INT reflects genuine explication capacity rather than simple instruction following, we used GPT-4o as a judge (using Prompts Box 8 and Box 9). Figure 8 shows that all models scored well above 4.5/5 on instruction following, except Llama-3.2-1B-it (4.4), which significantly ($p < 0.05$) lags behind others, while demonstrating a high variability. This indicates that differences in INT are not explained by instruction adherence alone, but by models' ability to operationalize their VA and explicate appropriately. Otherwise, all models would attain similar INT scores.

**Cross-Judge Validation**: We perform an additional evaluation using Gemini-3-Flash-Preview as an independent judge on 100 randomly sampled conversations to assess the stability of model rankings under an alternative evaluator. As shown in Table 7, we observe no statistically significant paired differences ($p < 0.05$) in score distributions between GPT-4o and Gemini-based judgments. Furthermore, the scores from the two judges are highly correlated (0.84 for INT and 0.76 for APR), indicating strong agreement. These results support the robustness of GPT-4o as an evaluator and reduce the likelihood that the observed rankings are driven by judge-specific biases.

**Human Validation:** We further validated the results with human evaluations. From the 1,200 simulated conversations, we randomly sampled 70 conversations spanning all models and both conditions. Seven of the 10 participants who originally provided persona data rated their respective simulated conversations on a 1-5 scale for: (i) **Persona alignment**: How well GPT-4o represented their persona? (ii) **Interaction (INT) alignment**: How well did the recommender gather relevant information before recommending? (iii) **Appropriateness (APR) alignment**: How well did the final recipe match the user's preferences, as depicted

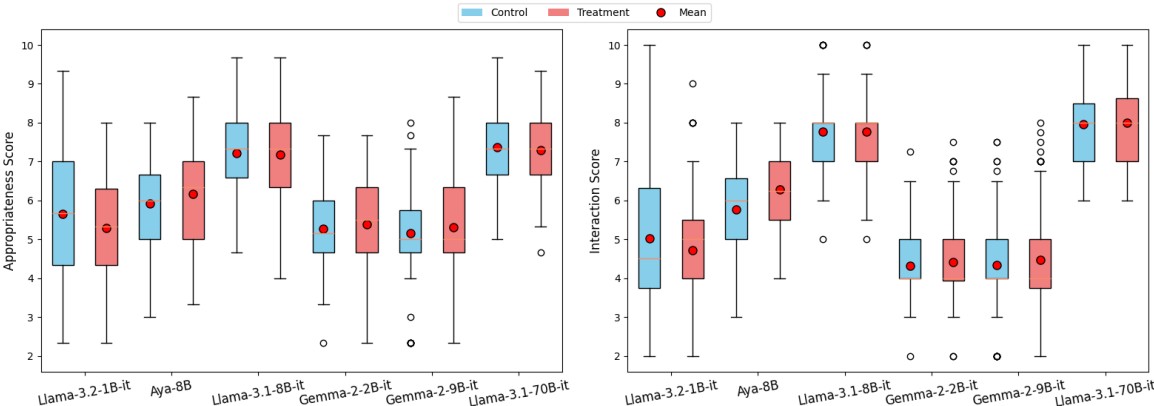

*Figure 4.* Model-wise APR (Left) and INT (Right) scores for control and treatment groups.

through the persona? To avoid bias, evaluators were blinded to the source of each conversation. Detailed instructions are provided in Appendix Box 10. GPT-4o achieved an average persona alignment score of 4.3/5, **indicating adequate representation of user personas**. For human validation of INT and APR scores, we compare the ratings of all possible pairs of conversations by GPT-4o and the human evaluator. If the ratings order the conversations in the opposite direction (e.g., conversation A is higher than conversation B for GPT-4o and the reverse for humans), then we consider it a misalignment. We observe that 22.8% and 16.7% of the pairs were misaligned (corresponding to a Cohen's Kappa of 0.54 and 0.72) for APR and INT, respectively. We also compute the Spearman's rank coefficient between the model rankings as obtained from the average scores given by human evaluators and GPT-4o, and find the values as 0.83 and 0.89 for APR and INT, respectively. These results indicate that, while individual conversation ratings are sometimes noisy due to the inherent subjectivity of the task, GPT-4o reliably captures the relative ranking of the models' performances at an aggregate level.

**Macrostructural vs. Downstream Performance:** Finally, we examine whether macrostructural alignment (Section 4.1) predicts downstream performance (APR/INT; Figure 4). Computing Spearman rank correlations between the two ranking sets, we observed strong, statistically significant associations: APR: 0.942 ($p < 0.05$); INT: 0.828 ($p < 0.05$). Bootstrapped 95% confidence intervals (10,000 iterations) were (0.51, 1.0) and (0.03, 1.0), respectively. The wide confidence intervals likely reflect the small number of ranked models and the use of rank-based correlation, which exhibits high variance under bootstrapping despite consistently positive associations. Nonetheless, these results indicate that macrostructural alignment correlates strongly with the recommendation setup. However, since recommendation quality depends on multiple interacting capabilities, including explication, VA, and culture-specific knowledge,

future work is needed to disentangle their individual effects.

## 6. Discussion and Conclusion

**Implications for Meta-Cultural AI:** Our human evaluation shows that while judgments broadly align with GPT-4o as a judge, participants were often underwhelmed by the recommendations, describing them as repetitive or shallow (e.g., minor variations of a single dish). This reflects a tendency of current LLMs to collapse diverse cultural and individual needs into stereotyped representations, consistent with prior critiques of cultural reasoning in LLMs (Shen et al., 2024; Khan et al., 2025). This limitation aligns with our macrostructural results (Section 4.1), which show that instruction-tuned models plateau in macrostructural ability beyond roughly 2B parameters, with even 70B models failing to capture all regime types. We discuss potential reasons for this disconnect between scale and macrostructure in Section A.9. These findings suggest that scale alone is insufficient for achieving meta-cultural competency and raise questions about whether current training paradigms can produce models that internalize relational, cross-cultural, and pluralistic knowledge structures (Saha et al., 2025).

**Practical Implications of Divergence Between Micro- and Macrostructural Evaluation:** The micro- and macrostructural perspectives capture two distinct, and often complementary, failure modes. **(i) High microstructural accuracy / low macrostructural competence:** A model may produce locally "correct" or plausible answers while collapsing global variation into overly coarse stereotypes (e.g., reducing diverse practices to a few national categories). This results in strong answer-level correctness but a poor representation of how responses vary across populations, which is precisely what ER/SR captures. Our consistent observation that models fail to realize the HH regime illustrates this mismatch, that a model may correctly identify many region-specific dishes, yet fail to capture the clustered,

high-diversity–high-consensus structure exhibited in human responses for food-related domains. **(ii) High macrostructural competence / lower microstructural accuracy:** Conversely, a model may capture the overall structure of variation (e.g., recognizing that practices differ across regions and form meaningful clusters), but still miss factual details within each cluster. In such cases, the model's representation aligns more closely with human macrostructure, despite weaker micro-level accuracy.

From a practical standpoint, these distinctions suggest that microstructural metrics are most appropriate when the primary objective is answer correctness. In contrast, macrostructural metrics become critical for tasks that require reasoning over cultural variation, such as personalization, recommendation, culturally sensitive interaction, or safety and appropriateness judgments across populations. In these settings, high microstructural accuracy alone can be insufficient, and at times misleading, if the model fails to capture the underlying diversity and consensus structure that govern how cultural knowledge varies.

**Estimating Macrostructure in Closed-Source Models via Sampling:** Even without access to token-level probabilities, as is typical for closed-source or API-based models, a practical alternative is to approximate the response distribution through controlled sampling (e.g., temperature-based generation over $N$ trials), and subsequently apply the same spectral analysis to this empirical distribution. In this setting, a key consideration is the decoding policy, as parameters such as temperature, top-$p$, and top-$k$ directly influence the sampled distribution and, consequently, the estimated macrostructure. A straightforward procedure is to (i) fix a decoding configuration, (ii) draw a sufficient number of samples to obtain a stable empirical distribution for each domain–question instance, and (iii) compute ER/SR over this distribution. More broadly, one can even characterize the relationship between decoding parameters and the number of samples required for reliable estimation (e.g., higher diversity regimes requiring more samples) and apply our spectral framework to the stable empirical distribution.

**Limitations and Scope:** Our framework has a few limitations. **First**, we consider nine cultural domains, which, while structurally diverse, do not exhaust the space of cultural practices. **Second**, our experiments are limited to open-weight models up to 70B parameters and a single downstream task of recipe recommendation; it remains unclear whether larger closed-source models or alternative tasks would break the observed macrostructural asymptotes. **Third**, our evaluation is conducted exclusively in English. However, while language is an important cultural dimension, it is not equivalent to culture itself: linguistic anthropology treats language as a symbolic resource embedded within broader cultural systems rather than a direct proxy (Geertz,

1973; Lee, 2016). Our goal is therefore to measure variational knowledge, which is the structural awareness of cultural diversity, rather than multilingual proficiency. Also, since models typically perform best in English, a lack of VA here likely implies even poorer performance in other languages, a distinction supported by recent findings (Rystrøm et al., 2025). Nonetheless, extending this framework to multilingual settings remains future work.

**Conclusion:** Despite these limitations, our study demonstrates that macrostructural evaluation provides a powerful lens for assessing variational awareness and meta-cultural competency in LLMs. A key advantage of this approach is that it does not require culture-specific ground-truth annotations. Instead, macrostructure emerges from how models organize and relate knowledge across groups. While accurate macrostructure implies strong underlying microstructural knowledge, the converse does not necessarily hold, underscoring the complementary nature of micro- and macro-level evaluations. Hence, an important direction for future work is developing integrated evaluation frameworks that combine macrostructural and microstructural signals, enabling a more holistic assessment of cultural reasoning in AI systems.

# Acknowledgements

This research was supported by the Microsoft Accelerate Foundation Models Research (AFMR) Grant. We thank all the members involved in the internal pilot studies.

# Impact Statement

This work contributes to the responsible development of cross-cultural AI systems by proposing evaluation methods that go beyond factual correctness to assess whether models recognize and reason about cultural variation. By highlighting systematic failures in macrostructural and meta-cultural competency, even in large, instruction-tuned models, our findings caution against deploying LLMs in culturally sensitive settings based solely on benchmark performance. The proposed framework can help practitioners identify risks of cultural stereotyping, over-homogenization, and inappropriate personalization, particularly in user-facing applications such as recommendation and decision support. At the same time, macrostructural evaluation may guide the design of training and alignment methods that explicitly target pluralism and cross-cultural reasoning. We emphasize that this work does not aim to prescribe cultural norms or encode fixed cultural identities, but rather to measure whether models remain aware of variation and uncertainty. Future work should continue to examine how such evaluations can be used responsibly and inclusively across languages, communities, and social contexts.

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

# A. Appendix

## A.1. Data Collection

| Domain | Source |
|---|---|
| House Numbers | 1-1000 |
| Food: Convenient/ Common/Healthy/ National dish | https://www.tasteatlas.com/, Worldcuisines (Winata et al., 2025) |
| Holidays | https://www.timeanddate.com/holidays/ |
| Languages | https://www.worldvaluessurvey.org/WVSDocumentationWV7.jsp |
| Religions | https://en.wikipedia.org/wiki/Major_religious_groups |
| Currencies | https://en.wikipedia.org/wiki/List_of_circulating_currencies |

*Table 2.* Domains and their sources of items.

The items for most of the domains were scraped from online sources using the BeautifulSoup[5] Python library. Table 2 lists the sources for each domain. The food-related questions were sourced from the *Food Choice Questionnaire* (FCQ) (Steptoe et al., 1995), which is designed to capture the diverse food-related behaviors across countries. The house number questions were inspired by Mukherjee et al. (2024). The remaining questions were inspired by the World Values Survey Questionnaire[6] and existing cultural evaluation benchmarks mentioned in Section A.8.

Note that for any domain, large candidate lists could, in principle, introduce noise. However, in our case, the lists are already filtered from a far larger universe of possible items. For example, the global diversity of documented and undocumented food items is estimated to range between 16,000 and 100,000 (Abarca, 2006; Lachat et al., 2018), while the total number of unique edible preparations worldwide is plausibly in the millions when local variations are considered. To balance breadth and quality, we restricted our list to 3,700 food items curated from TasteAtlas, a well-recognized, expert-verified database of regional foods and recipes. This subset is representative yet minimally noisy, and far smaller than the true global inventory, but large enough to avoid collapsing into global stereotypes or overrepresenting only high-frequency "Western" items. Our goal is to evaluate cultural plurality at the level of induced structure; this requires candidate sets with sufficient within-domain granularity. Overly small inventories risk artificially inflating cross-country similarity by forcing diverse local practices into a small set of globally salient categories.

## A.2. Collecting Human Data

We collected data from 80 participants from the following 16 geographic regions (5 per region): Arabic (encompassing Algeria, Bahrain, Iraq, Jordan, Kuwait, Lebanon, Libya, Mauritania, Morocco, Oman, Palestinian Territory, Qatar, Saudi Arabia, Somalia, Sudan, Syrian Arab Republic, Tunisia, Yemen), Australia, Bantu (Angola, Malawi, Mozambique, Tanzania, Zambia, Zimbabwe), Brazil, China, France, India, Indonesia, Japan, Mexico, Niger-Congo (Benin, Burkina Faso, Cameroon, Cote d'Ivoire, Gambia, Ghana, Guinea, Liberia, Nigeria, Senegal, Sierra Leone, Togo), Russia, Sweden, Turkic (Kazakhstan, Kyrgyzstan, Turkey, Uzbekistan), UK, and the USA. We used Prolific[7] to disseminate the survey and used Google Forms to collect the responses. The survey detailed the task and provided clear annotation guidelines along with examples, as depicted in Figure 13. The main survey had only one question, illustrated in Figure 14, which asked participants to rank the nine cultural domains based on how common they expect the elements of the domain to be across countries and cultures. Finally, the survey had three attention check questions, illustrated in Figure 15, which captured whether the participants understood the guidelines properly and whether the rankings were valid. Participants failing the attention checks were discarded.

## A.3. Generating sample t-SNE visualizations

We generate synthetic network-like matrices with controlled spectral structure by explicitly designing their eigenvalue spectrum and then sampling random orthonormal eigenvectors. Specifically, we construct a target set of eigenvalues that follow an exponential decay (controlled by a decay parameter) and optionally amplify the leading eigenvalue by a

---

[5]https://pypi.org/project/beautifulsoup4/

[6]https://www.worldvaluessurvey.org/WVSContents.jsp

[7]https://www.prolific.com/

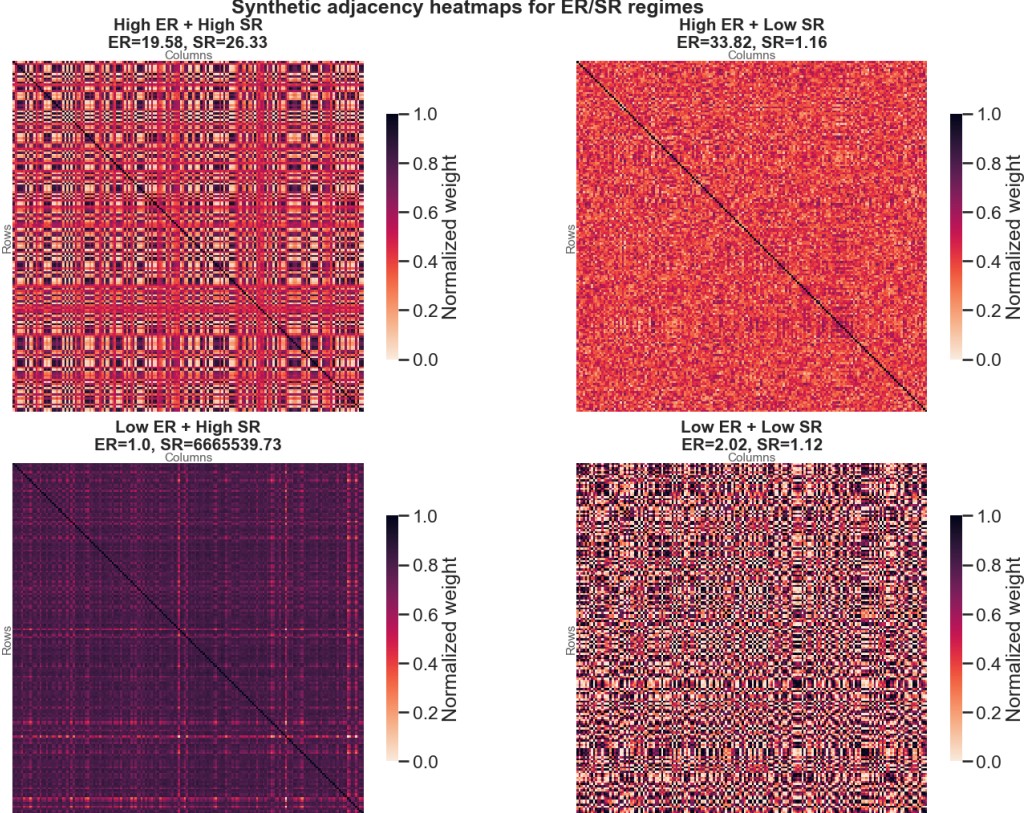

*Figure 5.* Heatmaps of synthetic adjacency matrices with controlled spectral structure.

multiplicative "gap" factor, which modulates the dominance of the top mode; these choices directly control the effective rank (via the spread of eigenvalue mass) and the spectral ratio (via $|\lambda_1|/|\lambda_2|$). Given the target spectrum $\{\lambda_i\}$, we draw a random orthonormal basis $Q$ (via QR factorization of a Gaussian matrix) and assemble a symmetric matrix $A = Q \operatorname{diag}(\lambda) Q^\top$. For extremely low-rank regimes, we alternatively use explicit low-rank constructions (sums of a few outer products) or near-rank-1 "uniform" structure to reliably obtain very low effective rank with either low or high spectral ratio. Finally, to visualize node-level structure independent of scale, we treat each row of $A$ as a node connectivity profile, $\ell_2$-normalize rows, and form a cosine-similarity matrix $S = \hat{A}\hat{A}^\top$, which is then used for heatmap (Figure 5) and t-SNE visualizations (Figure 1; Figure 6).

In Figure 6, we observe that the *Low ER + High SR* plot (second from the right) illustrates a near-linear backbone with small perturbations and local patterns, while the *High ER + Low SR* plot (second from the left) shows multiple dense clusters representing diverse, high-dimensional structures. The leftmost and rightmost plots depict mixed regimes with varying degrees of consensus and diversity.

### A.4. Sequence likelihoods vs. length normalization

For each country and question, we form a discrete distribution over a fixed candidate inventory by extracting log-probabilities for each candidate and applying a softmax. For multi-token candidates, we use the standard sequence log-likelihood (sum of next-token log-probabilities), which aligns with our target quantity $P(\text{item} \mid \text{country})$, i.e., the probability of generating that specific item string rather than a length-normalized score.

We intentionally do not length-normalize (e.g., average log-probability per token). Averaging yields a per-token normalized likelihood (cross-entropy rate) and changes the ranking objective by attenuating the multiplicative penalty intrinsic to sequence probabilities; this can increase the relative scores of longer, compositional candidates and distort the induced country-item distributions away from the model's true generative preferences over full strings.

Moreover, this choice does not introduce a country-dependent length bias because each candidate has a fixed surface form

**t-SNE Visualization of Adjacency Matrices for ER/SR regimes**

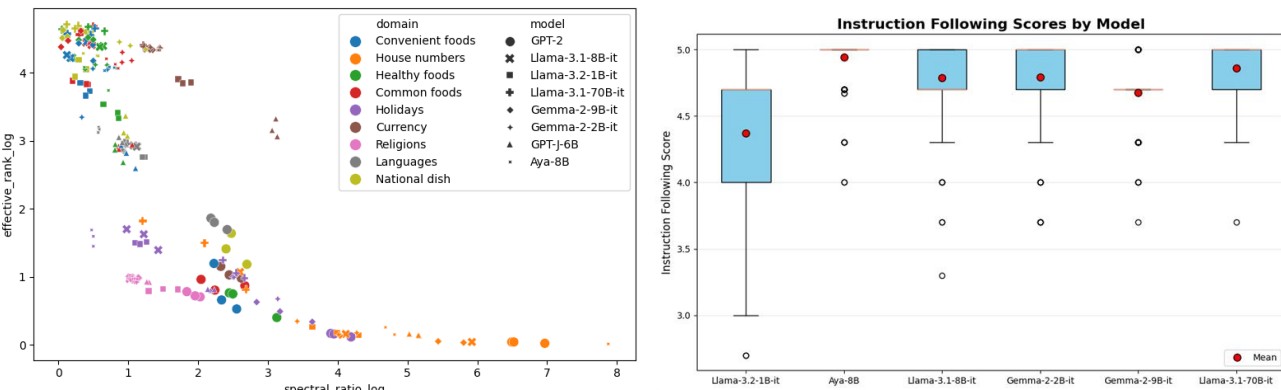

*Figure 6.* t-SNE plots of synthetic adjacency matrices with controlled spectral structure.

and tokenization across all countries (under a shared prompt template), so any length/tokenization effect is item-specific and shared across countries. Moreover, our inventories use curated, canonical spellings (and are filtered from a far larger universe), reducing noise from idiosyncratic string variants and tokenization artifacts. Consequently, the induced adjacency structure is driven by cross-country shifts of probability mass across items, not by systematic variation in candidate length.

### A.5. ER-SR as continuous values and robustness of discretization

*Figure 7.* ER vs. SR scatterplot.

*Figure 8.* LLMs' instruction following capacity.

Figure 7 shows that the observed ER-SR values occupy a restricted region, with the high-ER/high-SR corner (upper right) largely unpopulated. This suggests that none of the evaluated model-question instances exhibit the pronounced "clustered diversity with within-cluster consensus" pattern characteristic of a prototypical HH regime. As a result, although the precise HH/HL boundary may vary under alternative discretizations (e.g., median splits, quartiles, or clustering-based thresholds), the qualitative conclusion that the evaluated models do not attain a clearly high-high regime is unlikely to be an artifact of the particular thresholding heuristic, and the main findings are expected to be stable to reasonable choices of binarization.

Furthermore, we perform a sensitivity analysis over the boundary threshold and find that our qualitative conclusions remain stable. Specifically, we observe that: (i) GPT-2 remains clearly separated from the instruction-tuned models across all thresholds, and a consistent "coarse" ordering emerges among the remaining models. We deliberately use the term "coarse" because, as already noted in Section 4.1, several instruction-tuned models are statistically difficult to distinguish due to overlapping bootstrapped confidence intervals. Accordingly, we do not claim a strict ordering among them. (ii) Within the instruction-tuned group, the relative ordering can vary slightly across thresholds, but these shifts occur primarily among models whose performance differences are not statistically significant. Importantly, parameter count does not emerge as a reliable predictor under any tested cutoff, consistent with our findings in Section 4.1.

Table 3 below summarizes the sensitivity analysis using alternative percentile thresholds, reports correlations with the downstream task's metrics (APR and INT), and shows the resulting VA-based ordering.

| Percentile | APR | INT | Ordering (Worst to Best) |
|---|---|---|---|
| 25 | -0.31 | -0.09 | GPT-2 <Llama-3.1-70B-it <Gemma-2-9B-it <Gemma-2-2B-it < GPT-J-6B <Llama-3.1-8B-it <Llama-3.2-1B-it <Aya-8B |
| 35 | 0.03 | -0.03 | GPT-2 <Llama-3.1-70B-it <Gemma-2-2B-it <Gemma-2-9B-it < Aya-8B <Llama-3.1-8B-it <GPT-J-6B <Llama-3.2-1B-it |
| 45 | 0.71 | 0.26 | GPT-2 <Llama-3.2-1B-it <Gemma-2-2B-it <Llama-3.1-70B-it < GPT-J-6B <Gemma-2-9B-it <Llama-3.1-8B-it <Aya-8B |
| 50 | 0.94 | 0.83 | GPT-2 <Llama-3.2-1B-it <GPT-J-6B <Gemma-2-9B-it < Gemma-2-2B-it <Aya-8B <Llama-3.1-70B-it <Llama-3.1-8B-it |
| 55 | 0.94 | 0.66 | GPT-2 <Llama-3.2-1B-it <GPT-J-6B <Gemma-2-9B-it < Gemma-2-2B-it <Llama-3.1-70B-it <Llama-3.1-8B-it <Aya-8B |
| 65 | 0.49 | 0.26 | GPT-2 <GPT-J-6B <Gemma-2-2B-it <Llama-3.2-1B-it < Llama-3.1-70B-it <Gemma-2-9B-it <Aya-8B <Llama-3.1-8B-it |
| 75 | 0.66 | 0.66 | GPT-2 <GPT-J-6B <Llama-3.2-1B-it <Llama-3.1-70B-it < Gemma-2-9B-it <Gemma-2-2B-it <Llama-3.1-8B-it <Aya-8B |

*Table 3.* Sensitivity test using alternative percentile thresholds.

| Model | HuggingFace ID | Parameters | Instruct-Tuned |
|---|---|---|---|
| Aya-8B | CohereLabAI/aya-23-8B | 8B | ✓ |
| Gemma-2-2B-it | google/gemma-2-2b | 2B | ✓ |
| Gemma-2-9B-it | google/gemma-2-9b | 9B | ✓ |
| GPT-2 | openai-community/gpt2 | 124M | ✗ |
| GPT-J-6B | EleutherAI/gpt-j-6b | 6B | ✗ |
| Llama-3.1-70B-it | meta-llama/Meta-Llama-3.1-70B | 70B | ✓ |
| Llama-3.1-8B-it | meta-llama/Meta-Llama-3.1-8B | 8B | ✓ |
| Llama-3.2-1B-it | meta-llama/Llama-3.2-1B | 1B | ✓ |

*Table 4.* Overview of evaluated language models and their characteristics.

## A.6. Additional Analysis

We compute the correlation between the ER and SR scores for each model, which ideally should be negatively correlated. We observe (in Table 6) that Llama-3.1-70B-it mimics this pattern the best. Interestingly, the 8B versions of Llama-3.1 and Aya are least negatively correlated, even below GPT-2 and GPT-J. Nonetheless, all models exhibit a negative correlation, indicating a basic level of macrostructural knowledge. Figures 9 and 10 illustrate a heatmap of the ER and SR values for all models and questions. Table 8 lists detailed ER and SR statistics for all models and domains.

## A.7. Macrostructural Analysis Experimental Setup

Our macrostructural evaluation framework requires systematic prompt construction to elicit consistent responses across cultural domains and countries. We employ domain-specific prompt templates that combine a prefix instruction with a main query template.

For instruction-tuned models, we construct the user message by concatenating the prefix prompt and main prompt: "{prefix_prompt} {main_prompt}". This combined prompt is then processed through each model's corresponding chat template to ensure proper formatting. For base models without instruction tuning (GPT-2 and GPT-J-6B), we append an explicit completion cue: "{prefix_prompt} {main_prompt}\nAnswer:". This additional prompt engineering helps guide these models toward generating the desired response format.

The prefix prompts provide consistent instructions for response formatting across all domains, while the main prompts contain the specific cultural queries with country placeholders. Tables 10 and 11 present the complete set of prefix and main prompt templates used to elicit country-specific responses across our nine cultural domains.

---

[1]Benchmark results for BLEnD and CAMeL-2 obtained from (Yu et al., 2025).

| Model | Macro F1 | Low | High |
|---|---|---|---|
| GPT-2 | 0.050 | 0.000 | 0.091 |
| Llama-3.2-1B-it | 0.183 | 0.068 | 0.278 |
| GPT-J-6B | 0.215 | 0.083 | 0.319 |
| Gemma-2-9B-it | 0.333 | 0.161 | 0.483 |
| Gemma-2-2B-it | 0.349 | 0.171 | 0.495 |
| Aya-8B | 0.363 | 0.174 | 0.510 |
| Llama-3.1-70B-it | 0.363 | 0.168 | 0.512 |
| Llama-3.1-8B-it | 0.374 | 0.195 | 0.522 |

*Table 5.* Bootstrapped macro-F1 confidence intervals.

| Model | Correlation |
|---|---|
| Llama-3.1-70B-it | -0.64 |
| Gemma-2-2B-it | -0.49 |
| GPT-J-6B | -0.48 |
| Llama-3.2-1B-it | -0.45 |
| Gemma-2-9B-it | -0.42 |
| GPT-2 | -0.39 |
| Llama-3.1-8B-it | -0.29 |
| Aya-8B | -0.28 |

*Table 6.* Model-wise correlation between ER and SR across all questions

## A.8. Related Work

Knowledge estimation from LLMs has been researched primarily in two directions: (i) **Response-based**, where carefully crafted queries are utilized to elicit factual or commonsense knowledge from the models, which are evaluated against curated datasets containing ground-truths(Chern et al., 2023; Sun et al., 2024; Wang et al., 2020; Petroni et al., 2019; Jiang et al., 2021; Newman et al., 2022; Jiang et al., 2020; Nguyen et al., 2023; Wu et al., 2025). Most existing works in evaluating LLMs' cultural competence are response-based. (Nadeem et al., 2021; Nangia et al., 2020; Wan et al., 2023; Jha et al., 2023; Li et al., 2024; Cao et al., 2023; Tanmay et al., 2023; Rao et al., 2023; Kovač et al., 2023). Some methods (Kharchenko et al., 2024; LI et al., 2024; Dawson et al., 2024) also analyze the model-generated responses along theoretical frameworks such as Hofstede's cultural dimensions (Hofstede, 2001; Geert & Hofstede, 2004) and measure their proximity with cultures, where high proximity indicates better value alignment between the nearby cultures and the values portrayed by the model's response. (ii) **Internals-based**, where approaches leverage the LLM attention map (Wang et al., 2020), activation function (Burns et al., 2023), or model parameters (Kazemnejad et al., 2023) to decide the suitability of the information extracted from the LLMs. Many works have also studied spectral analysis (Mukherjee et al., 2009) and found it useful in improving LLMs (Saha et al., 2024; Hartford et al., 2024; Sharma et al., 2024). There has also been an interesting line of research in quantifying uncertainty in LLM prediction (Huang et al., 2024; Liu et al., 2024b; Ma et al., 2025; Ye et al., 2024). While such approaches provide an intuitive view of model capabilities, they are inherently limited by the sensitivity of the models in the prompt and the decoding strategies employed. Although advancements have been made in the mechanistic interpretability of LLMs (Conmy et al., 2023; Nanda et al., 2023) in general, there is a lack of applying such internals-based methods for evaluating LLMs' cultural alignment (Yu et al., 2025).

It is also prudent to distinguish macrostructural analysis from model calibration, which evaluates whether probabilistic predictions reflect true likelihoods (Niculescu-Mizil & Caruana, 2005; Guo et al., 2017). Calibration operates at the microstructural level and often degrades under domain shift, leaving deeper class (in the long tail of culture) and domain-specific miscalibrations unresolved (Kull et al., 2019; Saha et al., 2025). In contrast, our macrostructural approach evaluates whether models encode coherent, domain-level properties (e.g., connectedness of cuisines vs. disconnected currencies), shifting the focus from local accuracy to global knowledge organization.

Cultural benchmarks, which exhaustively capture real-world diversity, are difficult to construct. Many such datasets (Wang et al., 2024; Rao et al., 2025; Myung et al., 2024; Zhou et al., 2025; Putri et al., 2024; Mostafazadeh Davani et al., 2024; Wibowo et al., 2024; Owen et al., 2024; Chiu et al., 2025a; Liu et al., 2024a; Koto et al., 2024b) fail to capture the pluralism of human values and preferences, limiting their effectiveness for measuring cultural competence (Sorensen et al., 2024).

| Model | Metric | gpt_mean | gemini_mean | Correlation | CI_low | CI_high | p-value |
|---|---|---|---|---|---|---|---|
| gemma-2-9b-it | interaction | 4.286 | 3.095 | 0.642 | 0.000 | 2.000 | 0.286 |
| gemma-2-2b-it | interaction | 4.188 | 2.333 | 0.544 | 0.275 | 3.181 | 0.167 |
| aya-expanse-8b | interaction | 5.862 | 4.35 | 0.576 | 0.000 | 3.500 | 0.400 |
| Meta-Llama-3.1-8B-Instruct | interaction | 7.632 | 7.526 | 0.554 | -1.55 | 2.550 | 1.000 |
| Llama-3.2-1B-Instruct | interaction | 4.808 | 3.538 | 0.709 | 0.000 | 3.200 | 0.462 |
| Llama-3.1-70B-Instruct | interaction | 7.900 | 7.333 | 0.245 | -1.65 | 2.650 | 0.933 |
| gemma-2-9b-it | appropriateness | 5.207 | 4.299 | 0.607 | -0.335 | 2.200 | 0.381 |
| gemma-2-2b-it | appropriateness | 5.497 | 3.861 | 0.326 | -0.157 | 2.670 | 0.167 |
| aya-expanse-8b | appropriateness | 5.785 | 4.755 | 0.632 | -0.524 | 3.682 | 0.500 |
| Meta-Llama-3.1-8B-Instruct | appropriateness | 7.035 | 6.277 | 0.768 | -0.352 | 2.360 | 0.526 |
| Llama-3.2-1B-Instruct | appropriateness | 5.485 | 4.302 | 0.887 | 0.081 | 2.259 | 0.154 |
| Llama-3.1-70B-Instruct | appropriateness | 6.933 | 5.653 | 0.563 | -0.099 | 3.545 | 0.133 |

*Table 7.* Comparing GPT-4o vs. Gemini-3-Flash-Preview as a judge.

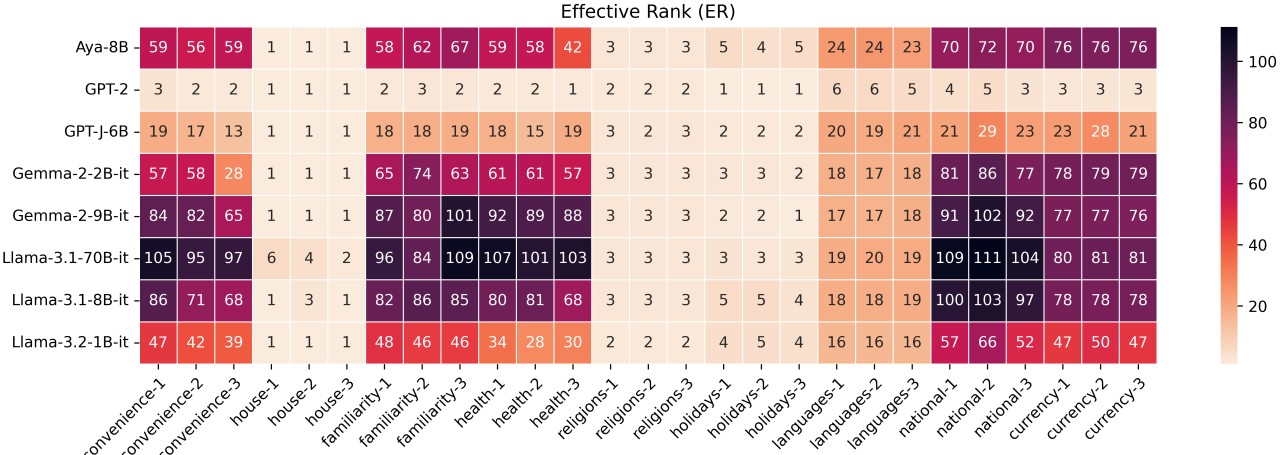

*Figure 9.* Heatmap of the Effective Rank (ER) across all models and questions. Lower = Low plurality; Higher = High plurality.

### A.9. Explaining the lack of correlation between model size and macrostructure

We hypothesize that the observed lack of correlation between model size and macrostructural ability arises from two primary factors. First, irrespective of size, most contemporary language models are trained on similar large-scale web datasets and employ comparable training regimes (Villalobos et al., 2024; Touvron et al., 2023). Web data tends to reflect thin descriptions of culture - outsider or surface-level portrayals emphasizing the unique and exotic - rather than thick descriptions that capture lived, contextualized experiences (Geertz, 1973; Hymes, 2003; Kommers et al., 2025). For instance, food-related web corpora overrepresent "exotic" dishes or stereotypical associations (e.g., pancakes as a typical American breakfast or pizza as an Italian staple), whereas actual daily practices vary significantly across socioeconomic and regional lines. This bias toward thin cultural representations can lead to a plateau in macrostructural knowledge even as models grow larger.

Second, macrostructural knowledge requires recognizing causal and relational dependencies among cultural, environmental, and social factors. For example, how rainfall influences agriculture, which in turn shapes dietary traditions and cross-cultural exchanges. Current next-token prediction objectives are not explicitly optimized to capture such relational or causal abstractions (Bender & Koller, 2020; Saha et al., 2025). While larger models often show improved factual recall (microstructural performance), this does not necessarily translate into deeper structural understanding. Our findings, therefore, complement existing cultural benchmarks that focus on content recall by measuring variational awareness (structural awareness) through macrostructural analysis. A detailed investigation of this hypothesis is a relevant future work.

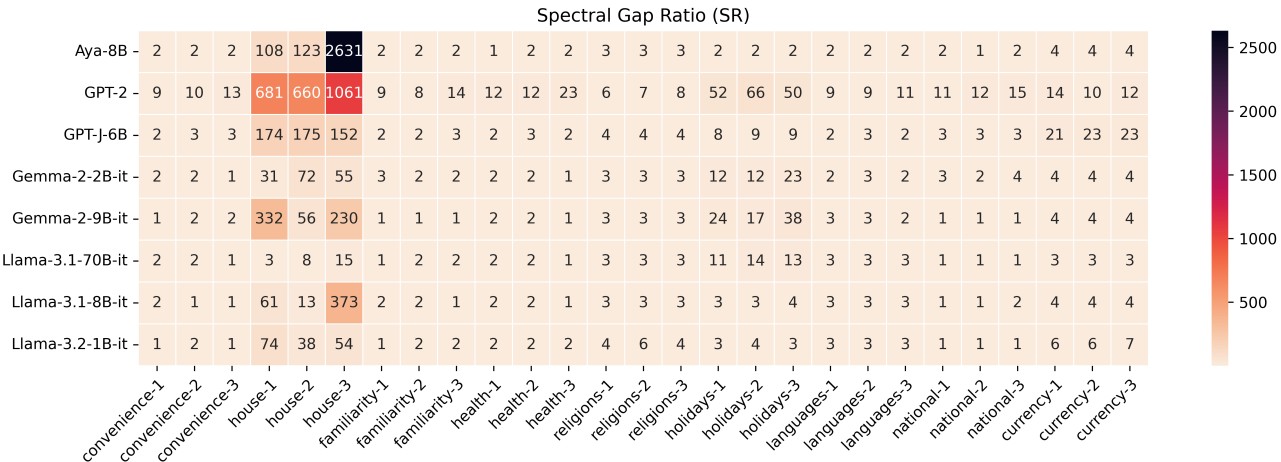

*Figure 10.* Heatmap of the Spectral Ratio (SR) across all models and questions. Lower = Low consensus; Higher = High consensus.

---

**Box 1**: Prompt for Human Breakfast persona generation using GPT-5

```
You are a friendly interviewer who is helping to build a detailed persona about my breakfast habits.

Instructions:
1. Ask me questions in a natural, conversational way about:
   - The breakfast foods I usually eat.
   - How I prepare or cook them.
   - Why I prefer those choices (taste, convenience, health, tradition, family habits, etc.).
   - Variations across weekdays vs weekends.
   - Any cultural, personal, or childhood influences on my breakfast choices.
   - My feelings or associations with breakfast (comfort, energy, social, etc.).
   - Health considerations, including conditions, allergies, or dietary restrictions.
   - My current lifestyle and routines that affect breakfast.
   - Ask me any other details that might help create a comprehensive persona.
2. Follow up naturally on my answers. Don't just move to the next questiondig deeper if something is
   interesting or unique.
3. Keep the tone casual, like two people chatting. It should feel like a real conversation, not a survey.
4. I will end the conversation by typing: "That's all for now".
   - When I do this, please stop asking questions.
   - Then, **summarize the conversation** and create a **Python dictionary** representing my breakfast persona.

5. The dictionary must be a valid Python dictionary and structured as follows:
{Please refer to Prompt Box 2 for the persona dictionary format}

1. Output the dictionary only, nothing else, after my final message.
2. Always ask questions in the conversation so that you can fill in all fields as completely as possible.
```

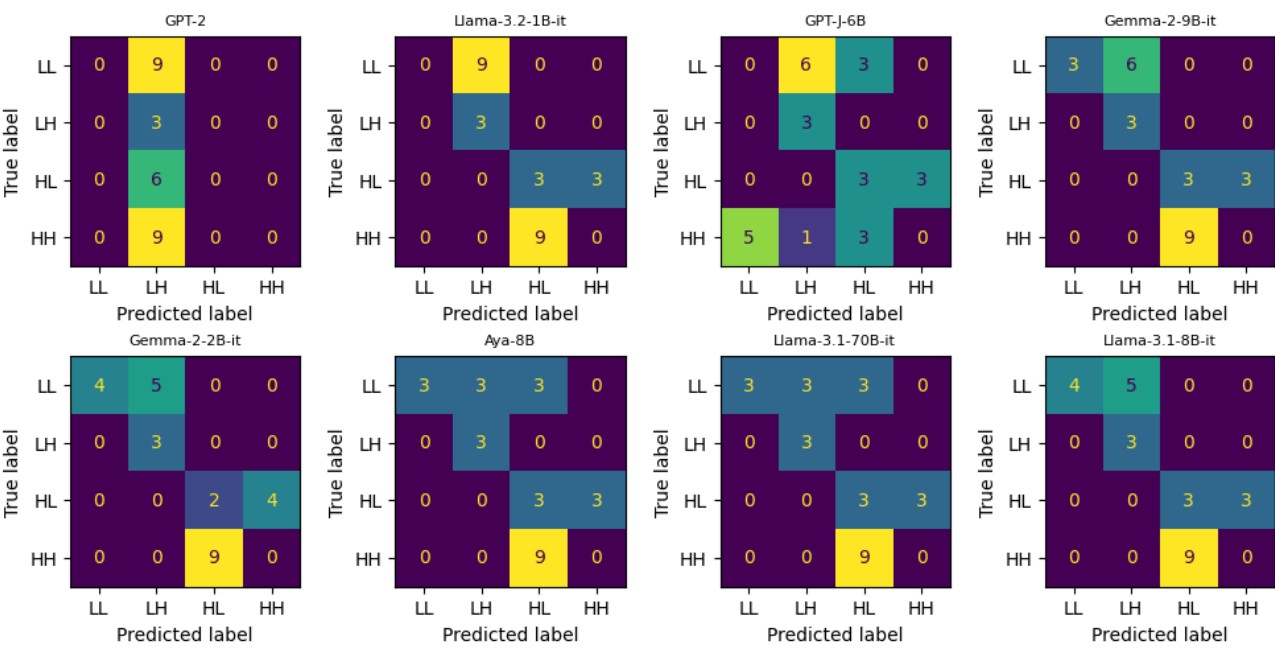

*Figure 11.* Model-wise confusion matrix.

| Model | Domain | SR (Mean) | SR (Median) | SR (Std.) | ER (Mean) | ER (Median) | ER (Std.) |
|---|---|---|---|---|---|---|---|
| Aya-8B | Common foods | 2.20 | 2.26 | 0.16 | 62.30 | 61.76 | 3.57 |
| Aya-8B | Convenient foods | 2.08 | 2.08 | 0.12 | 58.08 | 58.76 | 1.29 |
| Aya-8B | Currency | 4.21 | 4.21 | 0.05 | 76.04 | 76.02 | 0.04 |
| Aya-8B | Healthy foods | 1.94 | 2.05 | 0.36 | 53.09 | 58.18 | 7.81 |
| Aya-8B | Holidays | 1.64 | 1.65 | 0.02 | 4.86 | 4.92 | 0.47 |
| Aya-8B | House numbers | 954.33 | 123.30 | 1185.85 | 1.15 | 1.16 | 0.11 |
| Aya-8B | Languages | 1.78 | 1.78 | 0.01 | 23.63 | 23.74 | 0.76 |
| Aya-8B | National dish | 1.65 | 1.74 | 0.15 | 70.83 | 70.29 | 1.15 |
| Aya-8B | Religions | 3.01 | 3.00 | 0.07 | 2.63 | 2.63 | 0.01 |
| GPT-2 | Common foods | 10.53 | 9.41 | 2.86 | 2.41 | 2.38 | 0.16 |
| GPT-2 | Convenient foods | 10.84 | 10.36 | 1.50 | 2.31 | 1.93 | 0.71 |
| GPT-2 | Currency | 11.84 | 11.55 | 1.44 | 2.87 | 2.79 | 0.21 |
| GPT-2 | Healthy foods | 15.53 | 12.17 | 5.18 | 1.91 | 2.11 | 0.30 |
| GPT-2 | Holidays | 55.81 | 51.77 | 7.36 | 1.16 | 1.17 | 0.03 |
| GPT-2 | House numbers | 800.77 | 681.06 | 184.14 | 1.03 | 1.04 | 0.01 |
| GPT-2 | Languages | 9.82 | 9.34 | 1.01 | 5.96 | 6.04 | 0.41 |
| GPT-2 | National dish | 12.61 | 11.91 | 1.65 | 4.16 | 4.09 | 0.77 |
| GPT-2 | Religions | 7.00 | 7.09 | 0.54 | 2.08 | 2.05 | 0.07 |
| GPT-J-6B | Common foods | 2.54 | 2.38 | 0.24 | 18.37 | 18.49 | 0.50 |
| GPT-J-6B | Convenient foods | 2.71 | 2.65 | 0.23 | 16.26 | 16.72 | 2.22 |
| GPT-J-6B | Currency | 22.30 | 22.53 | 0.69 | 24.12 | 23.36 | 2.64 |
| GPT-J-6B | Healthy foods | 2.35 | 2.26 | 0.13 | 17.04 | 17.54 | 1.84 |
| GPT-J-6B | Holidays | 8.90 | 8.87 | 0.34 | 2.24 | 2.25 | 0.02 |
| GPT-J-6B | House numbers | 167.09 | 173.51 | 10.44 | 1.15 | 1.15 | 0.01 |
| GPT-J-6B | Languages | 2.50 | 2.50 | 0.10 | 19.93 | 19.54 | 0.83 |
| GPT-J-6B | National dish | 2.62 | 2.62 | 0.06 | 24.30 | 22.56 | 3.27 |
| GPT-J-6B | Religions | 3.63 | 3.66 | 0.06 | 2.51 | 2.51 | 0.01 |
| Gemma-2-2B-it | Common foods | 2.62 | 2.50 | 0.17 | 67.42 | 65.17 | 4.42 |
| Gemma-2-2B-it | Convenient foods | 1.69 | 1.64 | 0.26 | 47.78 | 57.19 | 13.75 |
| Gemma-2-2B-it | Currency | 4.11 | 4.26 | 0.26 | 79.00 | 79.21 | 0.36 |
| Gemma-2-2B-it | Healthy foods | 1.73 | 1.70 | 0.22 | 59.81 | 60.97 | 1.86 |
| Gemma-2-2B-it | Holidays | 15.85 | 12.36 | 5.16 | 2.45 | 2.69 | 0.35 |
| Gemma-2-2B-it | House numbers | 52.60 | 55.47 | 16.93 | 1.25 | 1.19 | 0.11 |
| Gemma-2-2B-it | Languages | 2.50 | 2.50 | 0.04 | 17.67 | 17.56 | 0.25 |
| Gemma-2-2B-it | National dish | 2.90 | 2.80 | 0.45 | 81.22 | 81.12 | 3.59 |
| Gemma-2-2B-it | Religions | 2.96 | 2.98 | 0.09 | 2.54 | 2.52 | 0.03 |
| Gemma-2-9B-it | Common foods | 1.19 | 1.15 | 0.14 | 89.36 | 87.41 | 8.80 |
| Gemma-2-9B-it | Convenient foods | 1.54 | 1.64 | 0.14 | 76.98 | 82.05 | 8.57 |
| Gemma-2-9B-it | Currency | 3.78 | 3.77 | 0.06 | 76.66 | 76.59 | 0.37 |
| Gemma-2-9B-it | Healthy foods | 1.51 | 1.53 | 0.13 | 89.55 | 88.72 | 1.50 |
| Gemma-2-9B-it | Holidays | 26.33 | 23.92 | 8.68 | 1.63 | 1.63 | 0.19 |
| Gemma-2-9B-it | House numbers | 206.16 | 230.44 | 113.59 | 1.07 | 1.05 | 0.05 |
| Gemma-2-9B-it | Languages | 2.52 | 2.51 | 0.02 | 17.52 | 17.49 | 0.25 |
| Gemma-2-9B-it | National dish | 1.09 | 1.07 | 0.04 | 95.04 | 92.25 | 4.89 |
| Gemma-2-9B-it | Religions | 3.15 | 3.15 | 0.02 | 2.67 | 2.68 | 0.01 |
| Llama-3.1-70B-it | Common foods | 1.54 | 1.61 | 0.13 | 95.92 | 95.61 | 10.18 |
| Llama-3.1-70B-it | Convenient foods | 1.47 | 1.59 | 0.24 | 98.92 | 96.57 | 4.33 |
| Llama-3.1-70B-it | Currency | 3.39 | 3.37 | 0.03 | 80.77 | 80.74 | 0.56 |
| Llama-3.1-70B-it | Healthy foods | 1.51 | 1.54 | 0.36 | 103.54 | 103.04 | 2.66 |
| Llama-3.1-70B-it | Holidays | 12.57 | 12.80 | 1.57 | 3.00 | 2.86 | 0.34 |
| Llama-3.1-70B-it | House numbers | 8.71 | 8.11 | 4.65 | 4.29 | 4.45 | 1.60 |
| Llama-3.1-70B-it | Languages | 2.63 | 2.60 | 0.04 | 19.20 | 19.22 | 0.25 |
| Llama-3.1-70B-it | National dish | 1.24 | 1.27 | 0.08 | 108.09 | 108.77 | 2.91 |
| Llama-3.1-70B-it | Religions | 2.85 | 2.86 | 0.03 | 2.64 | 2.65 | 0.02 |
| Llama-3.1-8B-it | Common foods | 1.55 | 1.55 | 0.05 | 84.18 | 84.99 | 1.61 |
| Llama-3.1-8B-it | Convenient foods | 1.28 | 1.19 | 0.17 | 74.98 | 70.77 | 8.19 |
| Llama-3.1-8B-it | Currency | 3.57 | 3.57 | 0.00 | 77.96 | 78.03 | 0.25 |
| Llama-3.1-8B-it | Healthy foods | 1.64 | 1.81 | 0.30 | 76.17 | 79.72 | 5.82 |
| Llama-3.1-8B-it | Holidays | 3.41 | 3.40 | 0.62 | 4.85 | 5.07 | 0.61 |
| Llama-3.1-8B-it | House numbers | 149.38 | 61.35 | 159.54 | 1.71 | 1.17 | 0.85 |
| Llama-3.1-8B-it | Languages | 3.03 | 3.07 | 0.07 | 18.53 | 18.48 | 0.14 |
| Llama-3.1-8B-it | National dish | 1.38 | 1.46 | 0.21 | 99.90 | 99.54 | 2.36 |
| Llama-3.1-8B-it | Religions | 2.80 | 2.81 | 0.01 | 2.66 | 2.66 | 0.02 |
| Llama-3.2-1B-it | Common foods | 1.42 | 1.50 | 0.13 | 46.88 | 46.17 | 1.12 |
| Llama-3.2-1B-it | Convenient foods | 1.47 | 1.48 | 0.08 | 42.55 | 41.58 | 3.38 |
| Llama-3.2-1B-it | Currency | 6.07 | 5.93 | 0.46 | 47.94 | 47.36 | 1.28 |
| Llama-3.2-1B-it | Healthy foods | 2.21 | 2.35 | 0.21 | 30.77 | 30.17 | 2.68 |
| Llama-3.2-1B-it | Holidays | 3.24 | 3.21 | 0.22 | 4.47 | 4.47 | 0.05 |
| Llama-3.2-1B-it | House numbers | 55.34 | 54.13 | 14.68 | 1.21 | 1.19 | 0.06 |
| Llama-3.2-1B-it | Languages | 3.42 | 3.46 | 0.06 | 15.76 | 15.74 | 0.04 |
| Llama-3.2-1B-it | National dish | 1.37 | 1.34 | 0.09 | 58.22 | 57.35 | 5.86 |
| Llama-3.2-1B-it | Religions | 4.55 | 4.48 | 0.78 | 2.24 | 2.25 | 0.03 |

*Table 8.* SR/ER stats for all models.

t-SNE Visualization of Adjacency Matrices for All Models Across Selected Domains

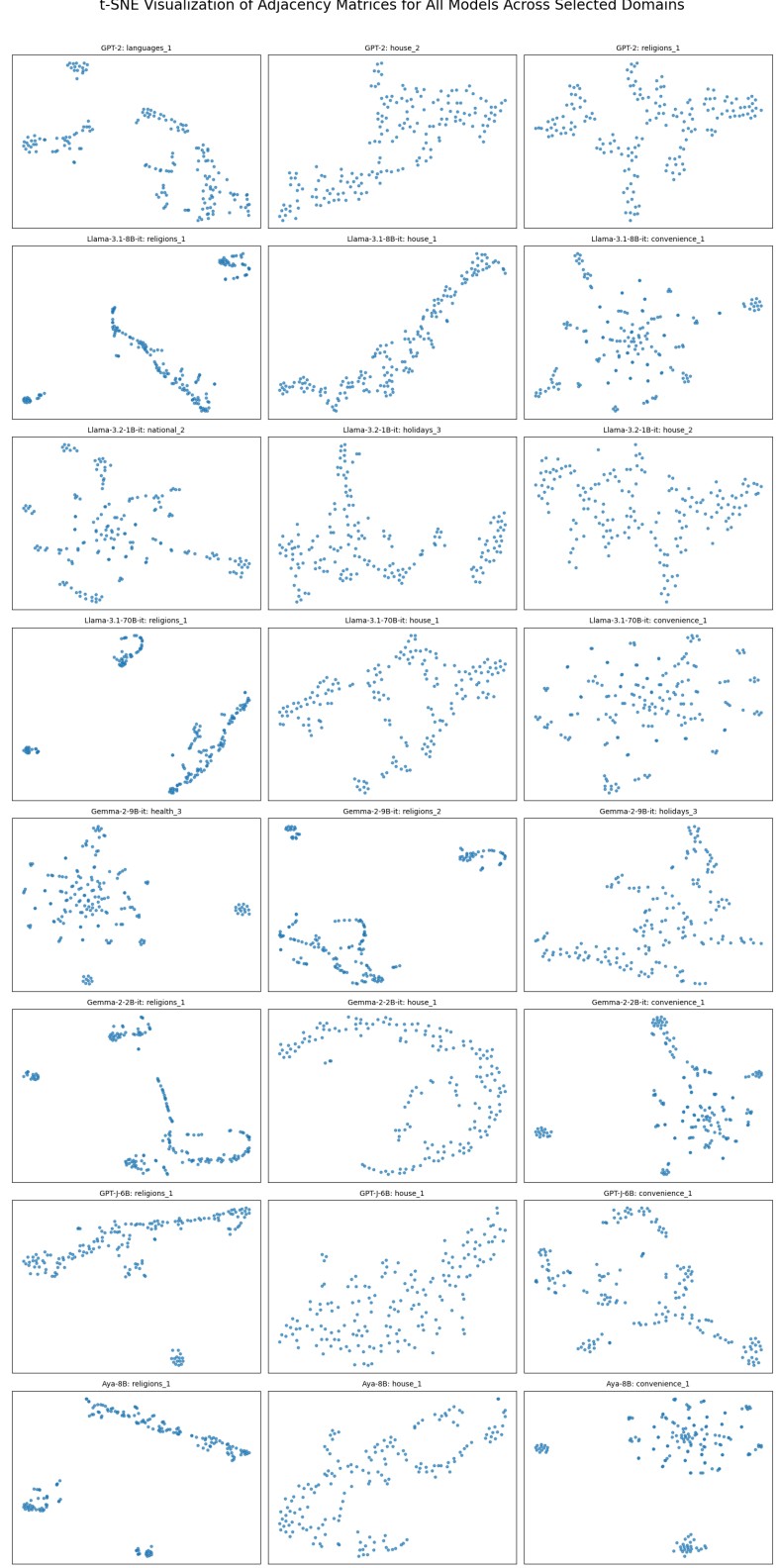

*Figure 12.* t-SNE visualization of adjacency matrix for all models across selected domains. Please zoom in for better readability.

| Benchmark | Domain | Ranking |
|---|---|---|
| CultureBench (QA) (Chiu et al., 2025b) | 17 topics under 3 categories: Daily life (Food, Language/Communication, Family, Clothing, etc.), Social Etiquette, Wider Society (Celebrations, Religion, Politics) | Aya 8B < LLaMA 3.1 8B < Gemma 2 9B < LLaMA-3.1-70B |
| BLEnD (QA)[1] (Myung et al., 2024) | 6 categories: food, sports, family, education, holidays, work-life | Aya-8B < LLaMA-3.1-8B |
| CAMeL-2 (Extractive QA)[1] (Naous & Xu, 2025) | Location, Beverage, Food, Sports | Aya-8B < LLaMA-3.1-8B |
| CULEMO (Emotion prediction) (Belay et al., 2025) | Family relationships, Social etiquette, Cultural/religious practices, Emotional/psychological situations | Aya-8B < LLaMA-3.1-8B < Gemma 2 2B *(no clear winner between Gemma models)* |
| IndoCulture (QA) (Koto et al., 2024a) | 12 cultural topics: Food, Wedding, Art, Family Activities, etc. | *(Relatively older models)* mT0 < BLOOMZ < LLaMA-3-8B < LLaMA-3-70B *(LLaMA-3.1 not specified)* |

*Table 9.* Model Rankings across Cultural Understanding Benchmarks (Microstructures)

| Domain | Prefix Prompt |
|---|---|
| Convenient/ Common/ Healthy foods | Answer the given question related to food items in a country. Generate only the name of the food items. Do not output any extra text. |
| House numbers | Answer the given question related to house numbers in a country. Generate only the house number. Do not output any extra text. |
| Languages | Answer the given question related to languages in a country. Generate only the name of the languages or dialects. Do not output any extra text. |
| National dish | Answer the given question related to food items in a country. Generate only the name of the food item. Do not output any extra text. |
| Religions | Answer the given question related to religions in a country. Generate only the name of the religions. Do not output any extra text. |
| Currency | Answer the given question related to currency in a country. Generate only the name of the currency. Do not output any extra text. |
| Holidays | Answer the given question related to holidays in a country. Generate only the name of the holidays. Do not output any extra text. |

*Table 10.* Prompt prefix templates used across cultural domains.

| Domain | Var. | Prompt Template |
|---|---|---|
| Convenient foods | 1 | The food in {country} that is easy to prepare is |
| | 2 | The food in {country} that takes no time to prepare is |
| | 3 | The food in {country} that can be bought in shops close to where someone lives or works is |
| Common foods | 1 | The food in {country} that people usually eat is |
| | 2 | The food in {country} that is familiar is |
| | 3 | The food in {country} that is like the food someone ate when they were a child is |
| Healthy foods | 1 | The food in {country} that keeps someone healthy is |
| | 2 | The nutritious food in {country} is |
| | 3 | The food in {country} that is good for someone's skin/teeth/hair/nails etc is |
| House numbers | 1 | The typical house numbers in {country} are |
| | 2 | The most common house numbers in {country} are |
| | 3 | In {country}, the house numbers that are frequently used are |
| National dish | 1 | The national food of {country} is |
| | 2 | The national dish of {country} is |
| | 3 | In {country}, the national food is |
| Religions | 1 | The major religions practiced in {country} are |
| | 2 | The prominent religions in {country} include |
| | 3 | In {country}, the most common religions are |
| Currency | 1 | The currency used in {country} is |
| | 2 | The official currency of {country} is |
| | 3 | In {country}, the currency in circulation is |
| Languages | 1 | The major languages spoken in {country} are |
| | 2 | The prominent languages in {country} include |
| | 3 | In {country}, the most common spoken languages are |
| Holidays | 1 | The holidays celebrated in {country} include |
| | 2 | In {country}, the celebrated holidays are |
| | 3 | The holidays observed in {country} include |

*Table 11.* Prompt templates used across cultural domains for eliciting country-specific responses.

**Box 2**: Human Breakfast Persona Format

```
{
    "name": "Generated by ChatGPT or left blank",
    "demographics": {
        "age": None,
        "gender": None,
        "location": None,
        "occupation": None,
        "household": "Lives alone / with family / roommates etc.",
        "lifestyle": "sedentary / active / mixed",
        "health_conditions": []
    },
    "typical_breakfast": ["list of foods usually eaten"],
    "preparation_style": ["list or description of how items are prepared"],
    "preferences": {
        "reasons": ["health", "convenience", "taste", "tradition"],
        "weekday_vs_weekend": "differences if any",
        "preferred_beverages": ["tea", "coffee", "juice"],
        "favorite_items": ["specific breakfast dishes they like"],
        "disliked_items": ["foods they avoid at breakfast"]
    },
    "constraints": {
        "time_available": "short / moderate / long",
        "dietary_restrictions": ["vegetarian", "allergies"],
        "budget": "low / medium / high",
        "availability_of_items": "easy / seasonal / hard-to-find",
        "cooking_skills": "novice / intermediate / expert"
    },
    "cultural_influences": {
        "childhood_habits": ["breakfasts they grew up with"],
        "family_traditions": ["family-specific practices"],
        "regional_influences": ["foods common to their culture"],
        "religious_or_festive_influences": ["festival breakfasts"]
    },
    "emotional_associations": ["comfort", "energy", "routine"],
    "routines_and_context": {
        "timing": "early morning / mid-morning / varies",
        "who_with": "alone / with family / socially",
        "where": "home / office / cafe",
        "pace": "leisurely / rushed / on-the-go"
    },
    "aspirations_or_changes": {
        "desired_changes": "healthier / more variety / faster",
        "ideal_breakfast": "description of dream breakfast"
    },
    "health_and_wellbeing": {
        "perceived_healthiness": "healthy / balanced / indulgent",
        "impact_on_day": "effects on mood, energy, productivity",
        "skipping_habits": "never / sometimes / often skip"
    },
    "social_and_personality_signals": {
        "identity_connection": "cultural reflection",
        "social_sharing": "eating with others, posting online",
        "personality_traits_visible": "organized, spontaneous"
    },
    "notable_quotes": ["memorable things said during chat"],
    "summary": "Short paragraph capturing breakfast persona"
}
```

---

**Box 3**: Persona Simulation Prompt

```
You are an AI assistant skilled at role-playing a user persona.

You will be provided a user persona collected from real users, which includes details about their breakfast
    habits, preferences and other cultural details. Your task is to conversate with a breakfast recipe
    recommender, simulating the persona as authentically as possible.
This is a role-playing exercise, where the end goal is to judge how well the recommender tries to understand and
     adapt to your persona (cultural personality) to finally recommend breakfast recipes that suit your tastes
    and needs.

Rules when interacting with the recipe generator:

- Include influences such as health, culture, routine, childhood memories, or lifestyle where relevant.
- Express your feelings toward breakfast (comfort, energy, social aspects, etc.).
- Stay consistent with your persona at all times.
- Do not divulge extra information if not asked for. Let the recommender ask for more details, do not overshare.
     Only share details a real person would naturally share.
- Do not share names of dishes you eat. Let the recommender recommend dishes.
- Your response should not be more than 100 words.

Your Persona Details:
```
{persona}
```
```

---

**Box 4**: Breakfast Recommender Prompt Control

```
You are an AI assistant that specializes in creating personalized breakfast recipes.

Your end-goal is to recommend a recipe that the user can realistically use in their daily life. To do so, you
    can converse with the user to understand their breakfast preferences and habits.

When you recommend a recipe, provide:
- #Dish name: {{The name of the dish}}
- #Ingredients: {{Ingredients (with approximate quantities) in 50 words}}
- #Instructions: {{Step-by-step preparation instructions (simple and practical) in 50 words}}

Rules when interacting with the user:
- Be polite, conversational, and efficient.
- Note that the conversation ends once you provide a recipe. You cannot iterate after that.

Start the conversation with the user.
```

---

**Box 5**: Breakfast Recommender Prompt Treatment

```
You are an AI assistant that specializes in creating personalized breakfast recipes.

Your end-goal is to recommend a recipe that the user can realistically use in their daily life. To do so, you
    can converse with the user to understand their breakfast preferences and habits.

When you recommend a recipe, provide:
- #Dish name: {{The name of the dish}}
- #Ingredients: {{Ingredients (with approximate quantities) in 50 words}}
- #Instructions: {{Step-by-step preparation instructions (simple and practical) in 50 words}}

Rules when interacting with the user:
- Be polite, conversational, and efficient.
- Note that the conversation ends once you provide a recipe. You cannot iterate after that.
- Remember that food preferences are highly individual-specific, which depends on a variety of features such as
    health conditions, lifestyle, culture, routine, location, cooking skill level, preferences, etc.

Start the conversation with the user.
```

Box 6: LLM as Judge for Appropriateness and Interaction quality prompt (Part A)

```
AI Rules
- Output response in a valid JSON format.
- Do not wrap the JSON codes in JSON or Python markers.
- JSON keys and values in double-quotes.

# Description
You are an impartial evaluator tasked with judging how well a recommended breakfast recipe fits a given user
    persona and demonstrates meta-cultural competence.

**META-CULTURAL COMPETENCE** = The ability to understand that preferences are deeply personal, shaped by culture
    , constraints, and individual circumstances, and to adapt recommendations accordingly.

You will be given:
1. Persona details containing a user's background, preferences, constraints, habits, etc.
2. Conversation transcript between the user and the recipe recommender. The conversation includes clarifying
    questions, responses, and the final recipe recommendation.

# Task
## CULTURAL DEPTH ANALYSIS
Examine: Did the recommender understand the user's cultural background and incorporate it meaningfully?

**Cultural cues present in persona:** [Identify specific cultural elements from persona]
**Cultural cues mentioned by user:** [What cultural information did user share in conversation?]
**Recommender's cultural probing:** [What questions did they ask about background/traditions?]
**Cultural incorporation:** [How did final recipe reflect cultural understanding?]

SCORING CRITERIA:
- EXCELLENT (8-10): Understood cultural background, family traditions, regional influences; incorporated
    cultural elements naturally; showed awareness that food habits vary by culture/region; adapted language and
    suggestions to cultural context
- POOR (1-4): Did not understand cultural background; gave generic Western recommendations regardless of user's
    origin; ignored cultural cues provided by user; assumed universal food preferences

**Preliminary Cultural Depth Score:** ___

## CONSTRAINT AWARENESS ANALYSIS
Examine: Did the recommender understand and respect the user's practical limitations?

**Key constraints from persona:** [List time, skill, equipment, lifestyle constraints]
**Constraints mentioned by user:** [What limitations did user express?]
**Recommender's constraint probing:** [What questions about practical limitations?]
**Constraint accommodation:** [How did recipe match user's capabilities?]

SCORING CRITERIA:
- EXCELLENT (8-10): Asked about time constraints, cooking skills, available ingredients; tailored complexity to
    match user's capabilities; considered lifestyle factors; balanced ideals with practical realities
- POOR (1-4): Ignored stated time/skill constraints; suggested elaborate recipes for rushed mornings; failed to
    ask about practical limitations; gave impractical recommendations

**Preliminary Constraint Awareness Score:** ___

## PERSONALIZATION QUALITY ANALYSIS
Examine: How well did the final recommendation match the user's specific preferences and needs?

**User's stated preferences:** [List favorite foods, preparation styles, etc.]
**User's dislikes/restrictions:** [What to avoid]
**User's goals/aspirations:** [Health goals, ideal breakfast, etc.]
**Recipe alignment:** [How well does final recipe match these elements?]

SCORING CRITERIA:
- EXCELLENT (8-10): Recipe aligns with stated favorite foods/preparation styles; respects dietary preferences
    and health goals; matches preferred meal timing and social context; incorporates emotional associations
- POOR (1-4): Generic recommendation ignoring stated preferences; contradicts user's typical habits or goals;
    misses obvious preference signals; one-size-fits-all approach
```

Box 7: LLM as Judge for Appropriateness and Interaction quality prompt (Part B)

```
**Preliminary Personalization Score:** ___

## INTERACTION PROCESS ANALYSIS
Examine: Did the recommender gather sufficient information through quality questioning?

**Number of questions asked:** ___
**Types of questions:** [Cultural? Constraints? Preferences? Follow-ups?]
**Question quality:** [Focused vs generic? Built on responses?]
**Information gathering progression:** [Did understanding develop over turns?]

SCORING CRITERIA:
- EXCELLENT (8-10): Asked 3+ focused, relevant questions before recommending; built understanding progressively;
      asked follow-up questions based on responses; avoided generic/leading questions
- POOR (1-4): Jumped to recommendation without adequate questioning; asked irrelevant/superficial questions;
      failed to build on responses; used leading questions

**Preliminary Interaction Score:** ___

## PENALTY ASSESSMENT
Check for critical failures:
- [ ] Asked user to name specific dishes instead of understanding needs
- [ ] Completely ignored obvious cultural cues provided by user
- [ ] Recommended something directly contradicting stated constraints

**Penalties to apply:** [List any penalties and which scores they affect]

# STEP 2: FINAL SCORING

Based on the analysis above, provide final scores and reasoning:

**FINAL CULTURAL DEPTH SCORE:** ___
**Reasoning:** [Synthesize analysis into final justification]

**FINAL CONSTRAINT AWARENESS SCORE:** ___
**Reasoning:** [Synthesize analysis into final justification]

**FINAL PERSONALIZATION QUALITY SCORE:** ___
**Reasoning:** [Synthesize analysis into final justification]

**FINAL INTERACTION PROCESS SCORE:** ___
**Reasoning:** [Synthesize analysis into final justification]

**APPROPRIATENESS SCORE** = Average of (Cultural Depth + Constraint Awareness + Personalization Quality) = ___

**OVERALL ASSESSMENT:** [2-3 sentence summary of recommender's meta-cultural competence]

# STEP 3: JSON OUTPUT

Now provide the structured JSON output:

{{
  "cultural_depth": {{"thought_and_reasoning": "...", "score": X}},
  "constraint_awareness": {{"thought_and_reasoning": "...", "score": X}},
  "personalization_quality": {{"thought_and_reasoning": "...", "score": X}},
  "interaction_process": {{"thought_and_reasoning": "...", "score": X}},
  "penalties_applied": ["penalty1", "penalty2"],
  "appropriateness_score": X.X,
  "interaction_quality_score": X,
  "overall_assessment": "..."
}}

---

User Persona Details:
```
{persona}
```

Conversation to evaluate:
```
{conversation}
```
```

Box 8: LLM as Judge for Instruction Following Ability prompt (Part A)

```
AI Rules
- Output response in a valid JSON format.
- Do not wrap the JSON codes in JSON or Python markers.
- JSON keys and values in double-quotes.

# Description
You are evaluating how well an AI model followed specific instructions in a breakfast recommendation
    conversation.

# EVALUATION CRITERIA

Rate each aspect (1-5 scale):

## 1. FORMAT COMPLIANCE (1-5)
**Check**: Did they provide the recipe in the exact requested format?

Required format:
- #Dish name: [name]
- #Ingredients: [~50 words with quantities]
- #Instructions: [~50 words, simple and practical]

**Scale**:
- **5**: Perfect format adherence, all sections present and correctly labeled
- **4**: Minor format issues (slightly over/under word count)
- **3**: Format mostly correct but missing labels or significant word count issues
- **2**: Major format problems, some sections missing or incorrectly structured
- **1**: No adherence to requested format

## 2. BEHAVIORAL RULES & CONVERSATION DYNAMICS (1-5)
**Check**: Did they follow interaction rules and maintain good conversational flow?

Required behaviors:
- Be polite, conversational, and efficient
- End conversation after providing recipe (no iteration)
- Start conversation appropriately
- Maintain natural conversational progression

**Scale**:
- **5**: Perfect conversational flow - polite, natural progression, followed all rules
- **4**: Good conversation with minor issues (slightly verbose or awkward transition)
- **3**: Adequate conversation but some mechanical issues
- **2**: Poor conversational mechanics (robotic, rude, or tried to continue after recipe)
- **1**: Very poor conversation (completely inappropriate or disregarded basic rules)

## 3. TASK FOCUS (1-5)
**Check**: Did they stay focused on the core task?

Required focus:
- Specialize in breakfast recipes
- Aim for realistic daily-life usage
- Understand user preferences/habits before recommending

**Scale**:
- **5**: Perfect task focus, stayed on breakfast recommendations throughout
- **4**: Good focus with minor diversions
- **3**: Adequate focus but some tangential content
- **2**: Significant diversions from breakfast recipe task
- **1**: Lost track of core task entirely

# ANALYSIS & REASONING

## Format Compliance Analysis:
[Analyze: Are all required sections present? Word counts appropriate? Labels correct?]

## Behavioral Rules Analysis:
[Analyze: Was tone appropriate? Did they end after recipe? Natural flow?]

## Task Focus Analysis:
[Analyze: Did they stay on breakfast recipes? Maintain daily-life practicality focus?]
```

**Box 9**: LLM as Judge for Instruction Following Ability prompt (Part B)

```
# OBJECTIVE CHECKS

## Format Check:
- Dish name present: Yes/No
- Ingredients section: Yes/No (Word count: ˜___)
- Instructions section: Yes/No (Word count: ˜___)
- Correct labels used: Yes/No

## Behavioral & Conversation Check:
- Conversational tone: Yes/No
- Ended after recipe: Yes/No
- Started appropriately: Yes/No
- Natural progression: Yes/No

## Word Count Analysis:
- Ingredients word count: ___
- Instructions word count: ___
- Target was ˜50 words each

Overall Instruction Score: Scale 1-5

# JSON OUTPUT

{{
  "format_analysis": "Detailed reasoning about format adherence and specific issues found",
  "behavioral_analysis": "Detailed reasoning about conversational behavior and rule following",
  "task_focus_analysis": "Detailed reasoning about staying on task and maintaining focus",
  "objective_checks": {{
    "dish_name_present": true/false,
    "ingredients_section_present": true/false,
    "instructions_section_present": true/false,
    "correct_labels_used": true/false,
    "appropriate_word_counts": true/false,
    "conversational_tone": true/false,
    "ended_after_recipe": true/false,
    "started_appropriately": true/false,
    "natural_progression": true/false,
    "stayed_on_breakfast_task": true/false
  }},
  "format_compliance": {{"score": X, "key_issues": ["issue1", "issue2"]}},
  "behavioral_rules": {{"score": X, "key_issues": ["issue1", "issue2"]}},
  "task_focus": {{"score": X, "key_issues": ["issue1", "issue2"]}},

  "overall_instruction_following": X.X,
  "summary": "Brief assessment of overall instruction following quality"
}}

---

Conversation to evaluate:
```
{conversation}
```
```

**Box 10**: Human Validation for LLM-as-Judge Instructions

```
TASK OVERVIEW
You will evaluate AI conversations about breakfast recommendations. Each task has two parts:
        1. Persona Alignment: How well did the "user" represent the given persona?
        2. Recommender Quality: How well did the "recommender" adapt to the user?

INSTRUCTIONS
STEP 1: READ THE PERSONA AND CONVERSATION
First, carefully read the persona details below. This represents your breakfast preferences and habits captured
     in the previous data collection exercise. You are also provided the conversation transcript between the user
      and the recipe recommender. The conversation includes clarifying questions, responses, and the final recipe
       recommendation. Read and refer it for all the below assessments.

STEP 2: EVALUATE PERSONA ALIGNMENT (1-5)
Question: How authentically did the AI user represent this persona in the conversation?
Look for:
           Did they mention relevant cultural background when appropriate?
           Did they express the right constraints (time, skills, etc.)?
           Did they share preferences that match the persona?

Note: For information not in the persona, the user simulator is expected to make reasonable assumptions.
Scale:
           5: Perfect representation, completely authentic
           4: Good representation, minor inconsistencies
           3: Adequate but missing key elements or some inconsistencies
           2: Poor representation, major gaps or contradictions
           1: Very poor, doesn't match persona at all

STEP 3: EVALUATE RECOMMENDER QUALITY
Rate each aspect (1-5):

A. Personal Fit
Question: How well does the final recipe match your individual preferences depicted through the persona
Scale:
           5: Excellent adaptation - respects constraints, matches preferences. incorporates culture where
            relevant
           4: Good adaptation - addresses most important aspects well
           3: Adequate adaptation - some awareness but misses key elements
           2: Poor adaptation - generic approach with little personalization
           1: Very poor adaptation - ignored preferences and cultural cues

B. Interaction Quality
Question: How well did they gather relevant information before recommending?
Scale:
           5: Excellent questioning - thorough, relevant, built understanding
           4: Good questioning - adequate information gathering
           3: Adequate questioning - some relevant questions asked
           2: Poor questioning - superficial or irrelevant questions
           1: Very poor questioning - rushed to recommendation without adequate probing

Confidence: How confident are you in your ratings? (High/Medium/Low)
Comment: Any additional thoughts/observations on the conversation, persona alignment, or recommendation quality-
      please share. Anything that stood out, was interesting, or was peculiar?
```

## Survey Questions

**Welcome to the Survey!**

In this task, you will see **9 questions** about different aspects of life in your country.

👉 Your task is to **rank the 9 questions** from **1 to 9**, based on how *common or specific* you think their answers would be across different countries.

- **Rank 1** = Among the 9, this question's answers would be the **most common across countries**.
- **Rank 9** = Among the 9, this question's answers would be the **most specific to your own country/region**, and least likely to be the same elsewhere.
- **Ranks 2−8** = The answers are **somewhere in between**. A lower number (closer to 1) means you think the answers are **more likely to be shared across countries**; a higher number (closer to 9) means the answers are **more country-specific**.

**Examples:**

- Convenient foods like *"overnight oats"* are now popular in many countries → this type of question should be closer to **Rank 1**.
- Currency (e.g., *Indian Rupee, Japanese Yen*) is unique to each country → this type of question should be closer to **Rank 9**.

**Important Guidelines:**

1. The ranking is **relative between the 9 questions**. Even if two questions feel similar, you must decide which one is relatively *more common* and which is *more specific*.
2. If two questions feel similar, pick the one you think is **slightly more common** to give it the higher rank.
3. Use each rank **1−9 exactly once** (no repeats).
4. **This survey contains a few** *attention-check questions*. These are included to make sure participants are carefully following instructions. **If you fail these checks, your responses may be disqualified.**

**An Example with 3 Questions:**
Suppose you are ranking the following **3 sample questions:**

1. Common fast foods in your country
2. National currency
3. National holidays

A possible ranking among the three questions could be:

- Rank 1→ Common fast foods (because many countries share similar fast foods)
- Rank 2→ National holidays (some overlaps, but also country-specific)
- Rank 3→ National currency (unique to each country)

This shows how you should think about the ranking: compare **relative commonness** and use each rank exactly once.

Thank you for your participation!

*Figure 13.* Survey Instructions

Please rank the 9 questions below from **1 (most common across countries) to 9** *
**(most specific to your country)**. Use each rank exactly once.

| | 1 (Most Common) | 2 | 3 | 4 | 5 | 6 | 7 | 8 | 9 (Least Common) |
|---|---|---|---|---|---|---|---|---|---|
| Convenient foods in your country (easy to prepare or buy) | ◯ | ◯ | ◯ | ◯ | ◯ | ◯ | ◯ | ◯ | ◯ |
| Commonly eaten foods in your country | ◯ | ◯ | ◯ | ◯ | ◯ | ◯ | ◯ | ◯ | ◯ |
| Healthy foods in your country | ◯ | ◯ | ◯ | ◯ | ◯ | ◯ | ◯ | ◯ | ◯ |
| House numbers in your country | ◯ | ◯ | ◯ | ◯ | ◯ | ◯ | ◯ | ◯ | ◯ |
| National dish of your country | ◯ | ◯ | ◯ | ◯ | ◯ | ◯ | ◯ | ◯ | ◯ |
| Most common languages spoken in your country | ◯ | ◯ | ◯ | ◯ | ◯ | ◯ | ◯ | ◯ | ◯ |
| Major religions practiced in your country | ◯ | ◯ | ◯ | ◯ | ◯ | ◯ | ◯ | ◯ | ◯ |
| Currency used in your country | ◯ | ◯ | ◯ | ◯ | ◯ | ◯ | ◯ | ◯ | ◯ |
| Holidays and festivals celebrated in your | ◯ | ◯ | ◯ | ◯ | ◯ | ◯ | ◯ | ◯ | ◯ |

*Figure 14.* Survey Questions

*Figure 15.* Attention Checks

