# OpenReview forum: "Measuring Meta-Cultural Competency: A Spectral Framework for LLM Knowledge Structures"
_ICML.cc/2026/Conference — ICML 2026 regular_

### Official Review · Reviewer_fr4H · 2026-03-06

**Soundness:** 4
**Presentation:** 4
**Significance:** 4
**Originality:** 4
**Overall Recommendation:** 5
**Confidence:** 5

**Summary:**

The paper proposes a spectral-analysis framework to evaluate whether LLMs capture the global macrostructure and diversity of cultural knowledge across countries

**Compliance With Llm Reviewing Policy:**

Affirmed.

**Final Justification:**

I believe this paper shows strong novelty, addressing an important yet underexplored aspect of cross-cultural understanding in LLMs, and therefore deserves acceptance.

I have carefully considered the valuable feedback from other reviewers and appreciate their insights; however, similar to my own comments, most of these concerns pertain to minor details. Fundamentally, I do not consider them to raise substantial issues or undermine the core contribution.

As such, I believe the paper merits acceptance. I have indicated the highest confidence in my evaluation and would like to reiterate my strong confidence in this work.

**Key Questions For Authors:**

- The conclusion that macrostructural competence may not scale with model size may be somewhat premature. If the goal is to systematically study scaling with model size, it would be advisable to include more larger models in the evaluation.

- The readability of Section 3 could be improved by providing some concrete examples and their corresponding distributions (possibly simplified ones). Computing ER and SR for these examples and explaining their physical interpretations would help readers better understand the metrics.

- The paper could benefit from discussing scenarios where logp cannot be accessed (e.g., closed-source models such as GPT). For example, one possible alternative would be to approximate it by running the model multiple times (e.g., N trials). Readers may be interested in understanding how the method would apply to closed-source but state-of-the-art models, even though this slightly differs from the authors’ original experimental setup.

- The discussion on the Micro vs. Macrostructural aspects could be further expanded, rather than being limited solely to rankings. A deeper analysis of these two perspectives would help clarify their broader implications.

**Limitations:**

Yes

**Strengths And Weaknesses:**

I believe the paper addresses the important problem of studying cultural aspects in LLMs, and the proposed macrostructural perspective is quite novel. The paper is also very well written, with a clear motivation. Although the experimental scale is not extremely large, the results already provide several useful takeaways. Overall, the paper is of high quality. For the limitations and suggestions, please see the questions section for more details.

---

> ### Author Rebuttal · Authors · 2026-03-29
>
> We are glad that you enjoyed reading our paper and found it novel. We thank you for your meticulous and constructive review. Please see our responses below:
>
> ```(Q1)```: **We agree that a definitive scaling law claim would require a broader sweep of large models. However, that is not the primary objective of this paper.** As stated in the introduction lines 46 (right column) - 79 (left column), our objective in this paper was to introduce and validate a spectral-analysis-based framework for evaluating the macrostructures of cultural knowledge in LLMs, not to systematically characterize scaling with parameter count.
> Accordingly, in Section 4.1 (lines 242-245, right column), we present the observation that instruction-tuned models appear to plateau in macrostructural ability beyond a certain size as a conjecture based on the models evaluated. We do not claim this as a generalizable scaling result, validating which would require a dedicated study with more models and controlled training differences.
> As stated in lines 49 (right column) - 60 (left column), we evaluate eight varying-sized models across nine culturally diverse domains spanning all five dimensions in Newmark’s taxonomy (Material, Ecology, Social, Habit, and Customs) and covering 170 countries to establish our framework. Our main conclusion (lines 423 - 437, Section 6, right column) is that macrostructural evaluation provides a powerful lens for assessing variational awareness and downstream meta-cultural competency. This conclusion is independent of any strong claims about scaling.
>
> ```(Q2)```: We appreciate this suggestion and agree that concrete, simplified examples would improve the accessibility of Section 3, especially for readers without a technical background. **While we already provide examples in Section 3.1.1, along with additional intuition in Appendix A.3 and Figure 6, we will incorporate a small set of illustrative toy examples directly into Section 3**. Specifically, we will show example answer distributions corresponding to each of the four regimes, compute their ER and SR values, and briefly explain the resulting physical interpretations (diversity vs. consensus) to make the metrics more intuitive in the main text.
>
> ```(Q3)```: We agree this is an important practical consideration for applying our framework to closed-source models where token-level log probabilities/logits are not accessible (e.g., GPT-style APIs). **Even without logits, a feasible alternative is to recover an empirical response distribution via controlled sampling (e.g., temperature-based generation over N trials) and then run the same spectral analysis on the reconstructed distribution.** In this setting, the key additional requirement is to control the decoding policy, since parameters such as temperature, top-p, and top-k directly shape the sampled distribution and therefore the estimated macrostructure.
> Practically, one can (i) fix a decoding configuration, (ii) draw enough samples to obtain a stable empirical distribution for each domain-question instance, and (iii) compute ER/SR on that empirical distribution. More generally, it should be possible to characterize an empirical relationship between decoding settings and the sample size needed to reliably recover each regime (e.g., higher diversity regimes requiring more samples). With an additional page, we will add this discussion to the paper's limitations/discussion section.
>
> ```(Q4)```: We agree that the micro- vs. macrostructural discussion should go beyond ranking comparisons and more clearly articulate the broader implications of the two perspectives. **While we currently provide a deeper review of microstructural evaluation in Appendix A.8,** we will move and condense the most relevant portion into the main paper (space permitting) and expand the discussion to more directly contrast microstructural and macrostructural evaluation.
> Also, we agree that explicitly illustrating cases where macro and microstructures disagree would strengthen the discussion, and **we will add this analysis in the discussion section (Section 6) of the paper**, with an additional page. Conceptually, the two dimensions capture different failure modes: (i) high micro/low macro: Model gives locally plausible answers yet collapses global variation into coarse stereotypes, yielding a weak ER/SR structure. Our repeated missing-HH result is a clear example, indicating that models may know regional dishes but miss the clustered high-diversity–high-consensus structure humans show. (ii) high macros/low micro: Model captures the shape of variation (global clustering) but misses within-cluster (regional) facts. Practically, microstructural metrics suit tasks focused on answer correctness, while macrostructural metrics are critical when the downstream goal requires structural knowledge and reasoning over cultural variation, such as personalization, recommendation, safety judgments, etc.
>
> We hope our responses addressed your questions.

---

> > ### Author Rebuttal · Reviewer_fr4H · 2026-04-01
> >
> > Thank you to the authors for their proactive response. I believe my current score (accept) already reflects my evaluation of the paper, so I have decided to keep my rating

---

### Official Review · Reviewer_jBVn · 2026-03-12

**Soundness:** 2
**Presentation:** 3
**Significance:** 3
**Originality:** 3
**Overall Recommendation:** 3
**Confidence:** 3

**Summary:**

This paper proposes a spectral-analysis-based framework for evaluating the macrostructural cultural knowledge of large language models (LLMs). The central observation is that most existing cultural benchmarks assess only microstructural knowledge, testing whether LLMs associate specific countries with specific facts, but fail to capture how cultural knowledge is organized at a larger scale. The authors introduce two spectral metrics derived from country-country similarity matrices: Effective Rank (ER), which measures the diversity or pluralism of cultural patterns, and Spectral Ratio (SR), which measures the degree of cross-country consensus. Together, these metrics define four macrostructural regimes (LL, LH, HL, HH) that characterize how cultural knowledge is distributed globally. The framework is grounded in Cultural Consensus Theory and validated via a human study across nine cultural domains spanning all five of Newmark's cultural dimensions and 170 countries. Eight open-weight LLMs of varying sizes are evaluated. Key findings include: instruction-tuned models align more closely with human cultural structure than older or non-instruction-tuned models; macrostructural competence does not scale consistently with model size; and the proposed spectral metric (Variational Awareness) significantly predicts downstream meta-cultural performance in a food recommendation simulation task, outperforming microstructural benchmarks in this predictive role.

**Compliance With Llm Reviewing Policy:**

Affirmed.

**Key Questions For Authors:**

1. Threshold sensitivity for regime classification: The ER and SR values are binarized using median-based thresholds to assign domains to one of the four macrostructural regimes. How sensitive are the Variational Awareness (VA) scores and the downstream APR/INT correlations to the choice of threshold? For instance, would using different percentile cutoffs (e.g., 25th/75th) or continuous ER/SR values change the conclusions substantially? A sensitivity analysis here would strengthen confidence in the regime-based formulation.

2. Generalization of the downstream validation: The meta-cultural simulation is based entirely on a food recipe recommendation task. How confident are the authors that macrostructural alignment (VA) would similarly predict downstream meta-cultural performance in other domains (e.g., health, religious practices, legal norms)? Are there theoretical reasons or preliminary evidence suggesting the correlation would hold more broadly, or could food be a particularly well-structured domain for spectral analysis?

3. GPT-4o as both simulator and judge: The simulation relies on GPT-4o as the user simulator and also as the LLM judge for INT and APR scores. Does this introduce a self-reinforcing bias, potentially favoring models whose outputs are more stylistically or semantically similar to GPT-4o? If a different LLM were used as the judge, would the model rankings remain stable? An analysis of judge-model correlation or a cross-judge comparison would help address this concern.

4. Relationship between macrostructural and microstructural metrics: The paper argues that macrostructural evaluation provides complementary signals to microstructural benchmarks—high macrostructural competence does not guarantee high microstructural competence and vice versa. Could the authors provide more specific examples or a quantitative analysis illustrating cases where the two disagree (e.g., a model with high VA but low microstructural accuracy), and what this implies for how practitioners should prioritize these evaluation dimensions?

5. Scalability of the human grounding study: The regime categorizations used as ground truth rely on a relatively small human study (~10 Prolific participants for SR and 3 pilot participants for ER estimation per domain). Have the authors considered whether the regime labels are stable across larger or more culturally diverse annotator pools? How might annotator cultural backgrounds affect the grounding of these macrostructural regimes?

**Limitations:**

Yes. The authors have discussed their limitations clearly in Section 6, including: (1) the nine cultural domains do not exhaustively represent all cultural practices; (2) the study is limited to open-weight models up to 70B parameters and a single downstream task; (3) the evaluation is conducted exclusively in English, which does not fully reflect cross-lingual cultural variation; and (4) the evaluation is built around a food recommendation scenario, leaving generalizability to other task types as future work. The paper acknowledges these limitations openly and in context, which is appropriate. No significant negative societal impact is identified beyond the general risks of deploying LLMs in culturally sensitive applications—and the paper's contribution is oriented toward better measurement of cultural alignment, which is a net positive for responsible AI development.

**Strengths And Weaknesses:**

Strengths:

Soundness: The spectral framework is mathematically grounded and draws on Cultural Consensus Theory as a theoretical basis. The two proposed metrics—Effective Rank (ER) and Spectral Ratio (SR)—are well-motivated and interpretable. The downstream simulation provides some empirical evidence linking Variational Awareness (VA) to meta-cultural performance (Spearman r=0.942 for APR).

Presentation: The paper is clearly structured and easy to follow. Figure 1 effectively summarizes the full evaluation pipeline. The use of numbered boxes for simulation prompts aids reproducibility, and the paper honestly enumerates its limitations in the conclusion section.

Significance: The high-level framing—that existing cultural benchmarks evaluate only microstructural factual accuracy, missing how cultural knowledge is organized globally—is an important and timely critique. The direction toward macrostructural evaluation could influence future benchmark design.

Originality: Applying spectral analysis to measure cultural knowledge macrostructure in LLMs is a creative combination of existing tools. The grounding in Cultural Consensus Theory differentiates it from purely engineering-focused evaluation works.

---

Weaknesses:

Soundness (major): Several methodological issues undermine confidence in the core claims. (1) The ER and SR values are binarized using median-based thresholds to assign the four macrostructural regimes, but no sensitivity analysis is provided—the VA scores and downstream correlations may be sensitive to this threshold choice. (2) The human grounding study used very few participants (approximately 3 per domain for ER estimation, ~10 for SR), which is insufficient to establish reliable ground truth for cross-cultural structural labels. (3) GPT-4o serves as both the user simulator and the automated judge (APR, INT scores), introducing a potential circular bias that favors models whose outputs align with GPT-4o's style—this is not adequately controlled for. (4) The downstream validation is limited to a single task (food recipe recommendation) with only six instruction-tuned models, making generalization claims premature.

Significance (moderate): The evaluation is conducted exclusively in English, which is in fundamental tension with the claim of measuring cultural knowledge—especially since cultural knowledge is often language-embedded. The finding that macrostructural competence plateaus beyond 2B parameters raises important questions that remain underexplored.

Originality (minor): The individual components (spectral analysis, CCT, LLM-as-judge simulation) are well-established; the novelty lies in their combination. The paper does not sufficiently distinguish itself from concurrent work on cultural alignment evaluation beyond framing.

---

> ### Author Rebuttal · Authors · 2026-03-29
>
> Thank you for your constructive review. Please see our responses below:
>
> ```(Q1)```: To clarify, our goal is to introduce a spectral framework for cultural macrostructures, where median ER/SR binarization is an operational choice. As explained in (lines 250-254 left, Sec. 3.3; Sec. A.5), ER-SR geometry is continuous and discretization only sets a reporting boundary. Nonetheless, **following your suggestion, we swept thresholds from 25-75 percentiles (step=10) and the conclusions stayed stable**: (i) GPT-2 remains separated from instruction-tuned models; Several instruction-tuned models are hard to distinguish (similar to lines 234-258 right, Sec. 4.1). (ii) Within instruction-tuned models, minor swaps occur mainly among statistically indistinguishable models, and parameter count is not predictive under any cutoff. We will add the table and insights to Sec. 4.1.
>
> ```(Q2)```: As explained in lines 320 (left) - 301 (right), **the downstream recipe recommendation task is inherently interdisciplinary and cross-cultural, drawing on nutrition, culture, economy, and geography**. It tests personalization, conversational reasoning, cultural sensitivity, and safety. That said, **we agree broader downstream tasks would strengthen generalization, and note this limitation explicitly in lines 404-408 (right)**, which we intend to do in the future.
>
> ```(Q3)```: We address judge bias in two ways. First, human validation (Sec. 5.2, lines 364-381 right) shows substantial agreement with GPT-4o and high rank agreement between human- vs GPT-based rankings. **Second, following your suggestion, a cross-judge check with Gemini-3-Flash-Preview on 100 conversations shows strong correlations with GPT-4o (0.84 INT, 0.76 APR) and no significant paired difference in score distributions**. We will add these new details to Sec. 5.2 (with a per-model table).
>
> ```(Q4)```: We agree and will add this analysis to Sec. 6. Macro vs micro capture different failures: (i) high micro/low macro: Model gives locally plausible answers yet collapses global variation into coarse stereotypes, yielding weak ER/SR structure. Our repeated missing-HH result is a clear example, indicating that models may know regional dishes but miss the clustered high-diversity–high-consensus structure humans show. (ii) high macros/low micro: Model captures the shape of variation (global clustering) but misses within-cluster (regional) facts. Practically, microstructural metrics suit tasks focused on answer correctness, while macrostructural metrics are critical when the downstream goal requires structural knowledge and reasoning over cultural variation, such as personalization, recommendation, safety judgments, etc.
>
> ```(Q5)```: We agree larger/diverse annotator pools matter, but provide three clarifications: (i) **Our grounding uses 80 participants (Sec 3.2), not 10**. (ii) Following CCT (Sec. 2), **we do not assume a single cultural "ground truth", but use agreement/disagreement patterns to estimate a meta-level domain property (expected diversity/consensus)**. Annotators are therefore not providing a canonical “correct” cultural answer. Instead, their responses are used to estimate ER/SR that characterize the response distribution, as summarized in Table 1. (iii) Since ER/SR are aggregate spectral quantities, **they require fewer samples than fine-grained micro-level labeling** (e.g., factual QA), and our goal is to distinguish broad regimes rather than recover the full distribution, as explained in Secs. 2, A.8.
> Also, **we mitigate background effects via demographic filters and attention checks (Sec. A.2)** and view the remaining variations as intrinsic to culture rather than noise.
>
> **```Weaknesses```:**
>
> ```Limited language```: We agree language matters, but, as noted in lines 408-422 (Sec. 6 right), language and culture are not equivalent. Since our goal is variational knowledge and not multilingual proficiency, although the prompts are in English, they cover globally diverse cultural content, letting us probe cross-cultural variation. If VA is missing in English (often models' strongest setting), it is unlikely to improve in other languages. Nonetheless, multilingual extension is a valuable future work.
>
> ```Parameter plateau```: We agree this is interesting, but the paper's goal is to establish and validate the macrostructural framework, not explain scaling mechanisms. We discuss two plausible reasons for the scale-macrostructure disconnect in Sec. A.9, and can elaborate if the reviewer has specific questions.
>
> ```Originality```: While the components are known individually, our contribution (Intro, lines 40-60 left) is the framework-level advance: a white-box macrostructural evaluation of cultural knowledge via agreement/diversity structure, which prior cultural benchmarks (accuracy/preference-based) do not capture. We compare against microstructural approaches in Appendix A.8 and will move it to the main paper with an additional page.

---

> > ### Author Rebuttal · Reviewer_jBVn · 2026-04-06
> >
> > I thank the authors for their detailed and thoughtful rebuttal.
> >
> > On Q5 (Human Grounding Study): I appreciate the clarification that 80 participants were used for the SR ranking survey, not 10 as I initially interpreted.
> >
> > On Q2 (Downstream Generalization): The response reasonably defends recipe recommendation as an interdisciplinary testbed but largely restates existing limitations. I accept this as a valid future-work item rather than a critical gap, though the single-task constraint continues to limit the strength of generalization claims.
> >
> > On Q4 (Macro vs. Micro Disagreements): The proposed taxonomy of failure modes — high micro/low macro (stereotypical collapse) versus high macro/low micro (structural awareness without regional detail) — is a useful conceptual contribution. The connection to the missing-HH result is well articulated. I look forward to seeing this analysis in the revised Section 6.
> >
> > New Concern: Measurement Validity Across the Model Range
> > Upon closer examination of the paper's experimental results (particularly Figures 9–11 and Table 6), a more fundamental concern has emerged regarding the construct validity of the spectral framework when applied to low-capacity models.
> > The authors evaluate eight models, but three of them — GPT-2 (124M), GPT-J-6B (non-instruct), and Llama-3.2-1B-it — exhibit behavior patterns that raise questions about whether the spectral metrics are measuring cultural macrostructure or merely reflecting insufficient model capacity to condition on the prompt.
> >
> > Specific Evidence
> > (1) GPT-2 (124M, non-instruct): As a 124M-parameter base model without instruction tuning, GPT-2 likely lacks the capacity to meaningfully condition its output distribution on country names embedded in the prompt. The resulting spectral values may therefore reflect generic language model priors rather than any cultural signal, whether present or absent.
> > (2) GPT-J-6B (non-instruct): Predicts only HL and LH categories, never identifying LL or HH regimes (Figure 11). Food-domain ER values (13–19) are substantially below those of instruction-tuned models (47–105+). As a non-instruct model relying on a simple "Answer:" completion cue (Appendix A.7), the degree to which it produces meaningfully differentiated conditional distributions across 170 countries is uncertain.
> > (3) Llama-3.2-1B-it (instruct-tuned): Displays the same prediction pattern as GPT-J,  only HL and LH, despite being instruction-tuned. Food-domain ER values (30–47) fall well below those of 8B+ models. The authors themselves acknowledge its significantly lower instruction-following score (4.4 vs. 4.5+ for others, p < 0.05) and describe it as exhibiting "possible lack of following instructions and limited meta-cultural competency" (Section 5.2).
> > - The Ambiguity: The core issue is that two competing explanations exist for the low spectral scores of these models, and the current analysis cannot distinguish between them:
> >
> > Interpretation A (favors the framework): These models genuinely lack cultural macrostructural knowledge, and the framework correctly identifies this deficit. The low VA scores reflect a true absence of cultural variation awareness.
> > Interpretation B (challenges the framework): These models fall below the capacity threshold at which the spectral metrics yield interpretable cultural signals.
> >
> > Both interpretations are plausible, and distinguishing between them matters for the paper's claims.
> > Impact on Key Claims
> > "Macrostructural competence does not scale consistently with model size" (Section 4.1): If three of the eight models effectively fall below the measurement floor, the substantive comparison reduces to five models (Gemma-2-2B-it, Gemma-2-9B-it, Aya-8B, Llama-3.1-8B-it, Llama-3.1-70B-it). The scaling claim then rests on two observations from five models which constitutes thin evidence for a general conclusion about the relationship between model size and macrostructural competence.
> > GPT-2 as a meaningful baseline: The rebuttal's assertion that "GPT-2 remains separated" under all threshold choices is expected and uninformative if the separation reflects model capacity rather than cultural knowledge structure.
> >
> > Recommended Resolution
> > I would suggest a straightforward diagnostic that would resolve this ambiguity:
> > A random/permutation baseline. Replace country names with shuffled country names or random strings (e.g., "The food in XYZ123 that is easy to prepare is") and run the same pipeline. If a model's actual ER/SR values are not significantly different from this random baseline, it indicates the model is not conditioning on the country name, and its spectral values should not be interpreted as reflecting cultural macrostructure. This test can be applied to all models and would establish an empirical measurement floor, above which spectral values carry cultural signal, and below which they do not.

---

> > > ### Author Response · Authors · 2026-04-06
> > >
> > > Thank you for the careful follow-up.
> > >
> > > We agree that the ambiguity - whether a model has no macrostructural knowledge or is below a certain capacity - can matter in general. However, the suggested permutation/random-string baseline primarily measures whether a model meaningfully conditions on the country cue, which is a distinct research question and **has already been studied directly in prior work** (https://aclanthology.org/2024.emnlp-main.884.pdf
> > > ). In contrast, **our goal is to introduce a spectral framework for evaluating cultural macrostructures (Sec. 1), not to test whether model responses are sufficiently conditioned on the prompt**.
> > >
> > > As described in Sec. 3.3 and A.4, **we do not score the correctness of generated answers**. Instead, **we analyze logits over a fixed, globally curated set of plausible items, which substantially reduces reliance on instruction-following format** and isolates whether the model has learned the associations between the probe and the candidate set. If instruction-following were the dominant bottleneck, we would expect a clean monotonic ordering by capacity proxies (such as instruction tuning + parameter count). Instead, we observe that several instruction-tuned models are statistically hard to distinguish (Sec. 4.1, lines 234-258 right), and non-instruct GPT-J performs comparably to (and sometimes above) Llama-3.2-1B-it on macrostructural metrics.
> > >
> > > Accordingly, **weak prompt conditioning is not an unmodeled confound in our setting. Rather, it is one mechanism by which a model can fail to exhibit cultural macrostructure**. If a low-capacity model's distribution over the candidate set is largely invariant to the country cue (due to generic priors), ER/SR will reflect a low differentiated structure. This is an interpretable outcome under our construct, indicating the model does not encode/use country-conditioned variation over the tested item space.
> > >
> > > That said, we agree that claims about scaling should be stated carefully. Since our intent is not to propose a general capacity threshold, but to report what the framework reveals about the evaluated models, in the revision, we will tighten Sec. 4.1 (as mentioned in our response 1B to reviewer qn3G) to avoid over-generalization.
> > >
> > > We are glad that our earlier responses addressed the prior points, and we hope this clarifies the present concern.

---

### Official Review · Reviewer_qn3G · 2026-03-14

**Soundness:** 2
**Presentation:** 2
**Significance:** 3
**Originality:** 3
**Overall Recommendation:** 4
**Confidence:** 3

**Summary:**

The paper introduces a spectral framework for evaluating cultural knowledge in LLMs at the macrostructural level, arguing that existing benchmarks capture only pointwise factual associations while missing broader organizational patterns such as how many distinct cultural clusters exist and how much cross-national variation there is within a domain.
The authors construct country-country similarity matrices from model-induced probability distributions over cultural items. They evaluate models acorss 9 domains and 170 countries for macrostructural alignment and correlaton with downstream meta-cultural competency in recipe recommendation.

**Compliance With Llm Reviewing Policy:**

Affirmed.

**Key Questions For Authors:**

1. The simulation uses GPT-4o as both the user simulator and the judge. Have the authors tested whether the model rankings change under an alternative judge, and if not, how do they address the concern that the APR correlation reflects GPT-4o's own biases?

2. The high-diversity, high-consensus regime is nearly unpopulated across all evaluated models, which the authors acknowledge. What does this imply for the framework's ability to distinguish models on the dimension of cultural pluralism, which is its stated motivation?

**Limitations:**

yes

**Strengths And Weaknesses:**

### Strengths:

1. The micro versus macro distinction is an interesting conceptual contribution to cultural evaluation. That is a meaningfully different and harder question, and motivating it through spectral analysis rather than curated QA is a principled choice.

2. The paper includes human study of 80 participants from diverse regions and proposes a useful taxonomy of macrostructural regimes.

3. The simulation study is a good way to evaluate the cultural understanding of the models.

### Weaknesses:

1. The VA metric reduces to a four-class classification over nine domains, giving each model a macro-F1 score computed from at most 9 data points. The bootstrapped confidence intervals in Table 4 are wide enough that several model pairs are not clearly distinguishable. Although the conclusion drawn by the authors is that model size does not predict macrostructural performance, the data size is not too convincing.

2. Binarizing effective rank and spectral gap ratio at their medians is an operational choice without theoretical grounding. The paper acknowledges that the high-diversity, high-consensus regime is nearly unpopulated in the observed data, meaning the four-regime framework collapses in practice to three regimes and the boundary between the two high-diversity regimes is a bit arbitrary.

3. The downstream simulation study relies on 10 real participant personas and a GPT-4o user simulator. The Spearman correlation between macrostructural ranking and APR is computed over six models, so a rank correlation of 0.942 involves only six data points. At that sample size, a single rank inversion would substantially change the result. The claim that macrostructural alignment predicts meta-cultural competency is directionally plausible but not robustly established by these numbers.

---

> ### Author Rebuttal · Authors · 2026-03-28
>
> Thank you for your thoughtful and constructive review. We are glad that the micro-macro distinction and the spectral framing came across clearly, and that you found the human grounding and simulation validation useful. Please find our responses below:
>
> ``` 1 (A).  The VA metric reduces to a four-class ... at most 9 data points.```: We are sorry for the confusion here. **This comment rests on a misunderstanding of what is "learned from" the nine domains versus what is defined by the framework.** The four macrostructural regimes are induced by the two spectral quantities ER and SR, and are defined at the level of the ER-SR plane (Section 3.1.1). They are not computed from the nine domains. Rather, the domains are selected to span the regimes based on human grounding (Section 3.2/Table 1). Concretely, we first define the regime semantics (LL/LH/HL/HH) via ER and SR, then choose nine domains whose expected structures cover all regimes, and validate that coverage via human judgments (Table 1).
>
> **Also, the macro-F1 score for each model is computed from 27 data points and not 9.** As explained in lines 255-260 (left column, Section 3.3), for each of the 9 domains, we used 3 questions, resulting in 27 data points per domain, and not 9.
>
> ```1 (B). The bootstrapped confidence intervals in Table 4 ... the data size is not too convincing.```: Indeed, **our goal is not a strict total ordering, but to introduce the spectral-analysis-based framework for evaluating the macrostructures of cultural knowledge in LLMs** and report the empirical pattern it reveals. As noted (Sec. 4.1, lines 234-258 right), several instruction-tuned models are statistically hard to distinguish due to overlapping bootstrap CIs. This itself is consistent with the takeaway that macrostructural competence does not scale monotonically with parameter count in this setting.
> Nonetheless, to strengthen the presentation, **we additionally ran pairwise tests over bootstrapped macro-F1**, and found GPT-2 is significantly different from most instruction-tuned models, while differences among instruction-tuned models are often not significant. We will report these tests and frame results as coarse clusters (e.g., GPT-2 vs. instruction-tuned), rather than a strict ranking.
>
> ```2. Binarizing effective rank and ... regimes is a bit arbitrary.``` + ```(Q2) The high-diversity ... which is its stated motivation?```: We agree that binarizing using median is an operational choice, and **we state this explicitly in lines 251-254 (Sec. 3.3)** as a way to convert continuous ER/SR measurements into an interpretable classification task with standard metrics such as macro-F1. As explained in Section A.5, the ER-SR relationships to human judgments are continuous. Using quartile cutoffs, k-means clustering, or continuous regression would not change the ER-SR geometry, the correspondence to human judgments, or our qualitative findings. Figure 7 further shows that none of the evaluated model-question instances exhibit a prototypical HH pattern.
> Crucially, this does not mean the framework "collapses" to three regimes. **An "unpopulated HH" in model outputs is exactly a failure mode the framework is designed to reveal, not evidence that the four-regime ER/SR characterization (Sec. 3.1.1) is invalid.** Humans judge some domains, such as food, as HH, and we include multiple food domains as HH in Table 1. The fact that all evaluated models systematically misclassify these as HL is an empirical result: models tend to represent food as disconnected national categories rather than clustered regional/common structures (Sec. 4.1). Since HH is the most diagnostic regime where pluralism coexists with shared structure, models not realizing HH imply they do not capture this form of cultural pluralism. We will make this interpretation more explicit in Sec. 4.1 and the discussion.
>
> ```3. The downstream simulation study relies on 10 ... not robustly established by these numbers.```: As we already explained in lines 407-415 (Section 5.2), we agree that the rank correlation number should be treated as directional, since it is prone to variance due to the small number of ranked models. Nonetheless, they do indicate that macrostructural alignment correlates strongly with the recommendation setup.
>
> ```Q1. The simulation uses GPT-4o ... correlation reflects GPT-4o's own biases?``` We understand this concern, and we address it in two ways. **First**, we perform human validation (Sec. 5.2, lines 364-381 right), which shows substantial agreement with GPT-4o and high rank agreement between human- vs GPT-based model rankings.
> **Second**, following your suggestion, **we performed a cross-judge check with Gemini-3-Flash-Preview on 100 conversations**, and found strong correlations with GPT-4o scores (0.84 INT, 0.76 APR) and no statistically significant paired difference in score distributions. We will add these details to Sec. 5.2, with a full per-model table.
>
> We hope we addressed all your questions.

---

> > ### Author Rebuttal · Reviewer_qn3G · 2026-04-07
> >
> > Thank you for clearing up the ambiguities. The rebuttal has resolved some of the concerns but the remaining factors about the paper (strengths of correlation, validation size, etc.) remain as objective factors that weaken the rigor of the results. I will keep my score.

---

### Decision · Program_Chairs · 2026-04-30

**Decision:**

Accept (regular)

**Comment:**

The paper introduces a spectral framework for evaluating cultural knowledge in LLMs at the macrostructural level, arguing that existing benchmarks capture only pointwise factual associations while missing broader organizational patterns, such as how many distinct cultural clusters exist and how much cross-national variation there is within a domain. The authors construct country-country similarity matrices from model-induced probability distributions over cultural items.

In summary, the three reviewers’ opinions are divergent as WR (qn3G), WA (jBVn), and A (fr4H).

The major strengths are:

- Macrostructural evaluation for cultural evaluation is novel, interesting, significant and timely (qn3G, jBVn, fr4H)
- The spectral framework is mathematically grounded (jBVn)
- Inclusion of human study of 80 participants from diverse regions (qn3G)

The major common concerns are:

- Statistical insufficiency. The sample size is too small (jBVn, qn3G)
- Median-based thresholds for effective rank and spectral gap ratio are arbitrary. (jBVn, qn3G)
- Evaluation circularity. self-preference bias of GPT-4o. (jBVn, qn3G)
- Limited task and language. The study is conducted in English and is limited to a single food recipe task. (jBVn)

According to the rebuttal acknowledgement by reviewers, their concerns have been partially resolved. Though the sample size weakens the rigor of the results, this paper endows new perspectives on cultural evaluation, so I recommend WA.